# Computational modeling of cambium activity provides a regulatory framework for simulating radial plant growth

Ivan Lebovka[1], Bruno Hay Mele[2†], Xiaomin Liu[1], Alexandra Zakieva[1], Theresa Schlamp[1], Nial Rau Gursanscky[3], Roeland MH Merks[4,5], Ruth Großeholz[1,6]*, Thomas Greb[1]*

[1]Centre for Organismal Studies, Heidelberg University, Heidelberg, Germany; [2]Department of Agricultural Sciences, Università degli Studi di Napoli Federico II, Napoli, Italy; [3]Gregor Mendel Institute, Vienna Biocenter, Vienna, Austria; [4]Mathematical Institute, Leiden University, Leiden, Netherlands; [5]Institute of Biology, Leiden University, Leiden, Netherlands; [6]BioQuant, Heidelberg University, Heidelberg, Germany

**Abstract** Precise organization of growing structures is a fundamental process in developmental biology. In plants, radial growth is mediated by the cambium, a stem cell niche continuously producing wood (xylem) and bast (phloem) in a strictly bidirectional manner. While this process contributes large parts to terrestrial biomass, cambium dynamics eludes direct experimental access due to obstacles in live-cell imaging. Here, we present a cell-based computational model visualizing cambium activity and integrating the function of central cambium regulators. Performing iterative comparisons of plant and model anatomies, we conclude that the receptor-like kinase PXY and its ligand CLE41 are part of a minimal framework sufficient for instructing tissue organization. By integrating tissue-specific cell wall stiffness values, we moreover probe the influence of physical constraints on tissue geometry. Our model highlights the role of intercellular communication within the cambium and shows that a limited number of factors are sufficient to create radial growth by bidirectional tissue production.

## Editor's evaluation

The main contribution of the article is the establishment of a framework for radial plant growth that was constructed using experimental observations fed into elegant computational models. This study will be of interest to plant scientists and more generally developmental biologists, working on mechanisms of tissue growth patterning.

## Introduction

Stem cells in plants are crucial for their longevity and usually maintained in meristems, special cellular environments constituting protective niches (**Greb and Lohmann, 2016**). At key positions in the plant body, we find distinct types of meristems that maintain their activity throughout a plant's life cycle. Shoot and root apical meristems (SAM, RAM) are located at the tips of shoots and roots, respectively, driving longitudinal growth and the formation of primary tissue anatomy in these organs. Moreover, lateral meristems organized in cylindrical domains at the periphery of shoots and roots execute their thickening. The cambium is the most prominent among these lateral meristems (**Fischer et al., 2019**). Cambium cells are embedded in two distinct vascular tissues produced in opposite directions by

*For correspondence:
ruth.grosseholz@bioquant.uni-heidelberg.de (RG);
thomas.greb@cos.uni-heidelberg.de (TG)

Present address: †Department of Biology, University of Naples, Naples, Italy

Competing interest: The authors declare that no competing interests exist.

periclinal cell divisions: xylem (wood) and phloem (bast) (*Chiang and Greb, 2019*; *Haas et al., 2022*). These tissues carry out fundamental physiological functions: long-distance transport of water and nutrients in case of the xylem and translocation of sugars and a multitude of signaling molecules in the case of the phloem. Based on its tightly controlled bidirectionality of tissue production and resulting bipartite organization, the cambium is a paradigm for bifacial stem cell niches which produce two tissue types in opposite directions and are found across different kingdoms of life (*Shi et al., 2017*).

Balancing proliferation and differentiation within meristems is essential. In the SAM and the RAM, this balance is maintained via interaction between the pool of stem cells and the organizing center (OC) and the quiescent center (QC), respectively, where the rate of cell division is relatively low. Both domains form a niche within the meristem instructing surrounding stem cells via regulatory feed-back loops (*Sabatini et al., 2003*; *Sarkar et al., 2007*; *Mayer et al., 1998*; *Daum et al., 2014*; *Pi et al., 2015*). In comparison to apical meristems, functional characterization of cambium domains was performed only very recently. During their transition from stem cells to fully differentiated xylem cells, early xylem cells instruct radial patterning of the cambium, including stem cell activity and, thus, similar to the OC in the SAM, fulfill this role only transiently (*Smetana et al., 2019*). In addition to influence from the early xylem, phloem-derived DNA-BINDING ONE ZINC FINGER (DOF) transcription factors designated as PHLOEM EARLY DOFs (PEARs) move to cambium stem cells and stimulate their proliferation in a non-cell-autonomous manner (*Miyashima et al., 2019*). Furthermore, genetically encoded lineage-tracing experiments showed that cell divisions are mostly restricted to individual bifacial stem cells located in the central cambium feeding both xylem and phloem production (*Smetana et al., 2019*; *Shi et al., 2019*; *Bossinger and Spokevicius, 2018*). Altogether, these findings defined functional cambium domains and revealed some of their reciprocal communication.

Another central and well-established mechanism regulating cambium activity in the reference plant *Arabidopsis thaliana* and beyond (*Kucukoglu et al., 2017*; *Etchells and Turner, 2010*; *Etchells et al., 2015*; *Fisher and Turner, 2007*) is the action of a receptor-ligand pair formed by the plasma membrane-bound receptor-like kinase PHLOEM INTERCALATED WITH XYLEM (PXY), also known as TDIF RECEPTOR (TDR), and the secreted CLAVATA3/ESR-RELATED 41 (CLE41) and CLE44 peptides. Like the PEAR proteins (*Miyashima et al., 2019*), CLE41 and CLE44 are expressed in the phloem and thought to diffuse to dividing cells in the cambium area expressing PXY (*Etchells and Turner, 2010*; *Hirakawa et al., 2008*). Direct binding of CLE41 to PXY (*Hirakawa et al., 2008*; *Morita et al., 2016*; *Zhang et al., 2016*) promotes the expression of the transcription factor WUSCHEL RELATED HOMEOBOX 4 (WOX4) (*Hirakawa et al., 2010*), which, in turn, is crucial for maintaining the capacity of cells to proliferate (*Kucukoglu et al., 2017*; *Hirakawa et al., 2010*; *Suer et al., 2011*). At the same time, the PXY/CLE41 module is reported to repress xylem differentiation in a *WOX4*-independent manner (*Hirakawa et al., 2010*; *Kondo et al., 2014*). In this context, PXY stimulates the activity of glycogen synthase kinase 3 proteins (GSK3s), like BRASSINOSTEROID-INSENSITIVE 2 (BIN2) (*Kondo et al., 2014*). BIN2, in turn, represses the transcriptional regulator BRI1-EMS SUPPRESSOR 1 (BES1), which mediates brassinosteroid (BR) signaling and promotes xylem differentiation (*Kondo et al., 2014*; *Saito et al., 2018*). In line with the hypothesis of a dual role in regulating stem cell activity and xylem differentiation, *PXY* is expressed in the proximal cambium zone containing developing xylem cells and in the central cambium zone containing bifacial cambium stem cells (*Shi et al., 2019*; *Brackmann et al., 2018*; *Shi et al., 2020*).

Distally to the *PXY* expression domain and oriented towards the phloem, the closest homolog to PXY, the receptor-like kinase MORE LATERAL GROWTH 1 (MOL1), represses cambium activity (*Agusti et al., 2011b*; *Gursanscky et al., 2016*). Although their extracellular domains are highly similar, *PXY* and *MOL1* cannot functionally replace each other, indicating that MOL1 activity does not depend on CLE41/44 peptides and that distinct signaling loops act in the proximal and distal cambium domains (*Gursanscky et al., 2016*). The latter conclusion is supported by the finding that the AUXIN RESPONSE FACTOR5 (ARF5) is expressed in the proximal cambium and promotes the transition from stem cells to xylem cells by directly dampening *WOX4* activity (*Brackmann et al., 2018*; *Han et al., 2018*). ARF5 activity is enhanced by phosphorylation through the GSK3 BIN2-LIKE 1 (BIL1) which, in contrast to other GSK3s (*Kondo et al., 2014*), is inhibited by the PXY/CLE41 module (*Han et al., 2018*).

As the role of multiple communication cascades between different cambium-related tissues is beginning to emerge, it is vital to generate a systemic view on their combined impact on cambium

activity and patterning integrated into a dynamic tissue environment. However, although the cambium plays an instructive role for stem cell biology, a dynamic view on its activity is missing due to its inaccessibility for live-cell imaging. Computational modeling, in particular agent-based modeling combining tissue layout with biochemical signaling processes, can overcome these obstacles and help analyzing the interplay between cellular signaling processes, cell growth, and cell differentiation in silico that would otherwise be inaccessible. Here, we present a dynamic, agent-based computational model (*Merks and Glazier, 2005*) of the cambium integrating the functions of PXY, CLE41, and putative phloem-derived signals into a plant-specific modeling framework. As revealed by informative cambium markers, our model is able to reproduce anatomical features of the cambium in a dynamic manner. It also allows studying the cambium as a flexible system comprised of multiple interacting factors, and the effects of those factors on cell division, differentiation, and tissue patterning.

## Results

### Establishing a dynamic cambium model

Taking advantage of the almost exclusive radial expansion of mature plant growth axes, we sought to create a minimal framework recapitulating the 2D dynamics of radial plant growth. To do so, we first produced a simplified stereotypic 2D representation of a plant growth axis displaying a secondary anatomy by employing VirtualLeaf – a framework specially designed for agent-based modeling of plant tissue growth (*Merks et al., 2011*; *Antonovici et al., 2022*). To avoid confusion, we refer to factors within the model by an asterisk: for example, GENE– refers to the plant gene, whereas GENE* refers to its model counterpart. Within the model we defined three cell types: cells designated as cambium*, cells present in the center referred to as xylem*, and cells present distally to the cambium* designated as phloem* (*Figure 1A*). These cell* types were organized in concentric domains as observed after the establishment of a secondary organ anatomy (*Smetana et al., 2019*). To reduce the risk of losing cambium cells* during our simulations and allow differential cambium cell* behavior right from the start, we defined a rather large starting pool of cambium cells*. We then defined rules determining cell* behavior: (i) all cells* grew until they reached a size specific for each cell* type, (ii) cambium cells* divided when they exceeded a certain size, and (iii) cambium cells* changed their identity into xylem* or phloem* depending on the conditions described below (see also supporting information, *Supplementary file 1*). All chemical-like factors* implemented in the model had manually chosen cell* type-specific production and degradation rates.

To implement context-dependent regulation of cambial cell division and differentiation, we took advantage of the PXY/CLE41 signaling module (*Hirakawa et al., 2008*; *Hirakawa et al., 2010*): Phloem cells* produced a factor designated as CLE41* able to diffuse between cells*, whereas the corresponding, non-diffusing receptor designated as PXY* is produced in cambium cells* (*Figure 1B*). Recapitulating the CLE41-dependent function of PXY, we considered the following reaction:

$$CLE41 + PXY \rightarrow PXY_{active} \tag{1}$$

Thereby, the presence of both CLE41* and PXY* in a cell turned PXY* into $PXY_{active}$* (*Figure 1B*). For cambium cells*, we described the PXY*-CLE41* interaction by the following equations:

$$\frac{d}{dt}\left[PXY^*_{active}\right] = \left[PXY^*\right] \cdot \left[CLE41^*\right] - degradation_{PXY_{active}} \cdot \left[PXY^*_{active}\right] \tag{2}$$

$$\frac{d}{dt}\left[PXY^*\right] = \frac{production_{PXY}}{\left(1 + suppressrate \cdot \left[PXY^*_{active}\right]\right)} - \left[PXY^*\right] \cdot \left[CLE41^*\right] - degradation_{PXY} \cdot \left[PXY^*\right] \tag{3}$$

$$\frac{d}{dt}\left[CLE41^*\right] = diffusion_{CLE41} - \left[PXY^*\right] \cdot \left[CLE41^*\right] - degradation_{CLE41} \cdot \left[CLE41^*\right] \tag{4}$$

In these equations, [X*] denotes the concentration of the respective factor in each cell*. Since PXY-CLE41 signaling was reported to negatively regulate *PXY* expression (*Etchells and Turner, 2010*), we assumed that the production rate of PXY* is inhibited by [$PXY_{active}$*]. Therefore, the higher [$PXY_{active}$*] in a given cell*, the less PXY* was produced (*Equation 3*). To integrate PXY/CLE41-dependent regulation of cell proliferation, we let cambium cells* divide only when [$PXY_{active}$*] exceeded a certain threshold. Thereby, the proliferation of cambium cells* was dependent on both, locally produced PXY* and CLE41* originating from the phloem*. To instruct the differentiation of cambium cells*,

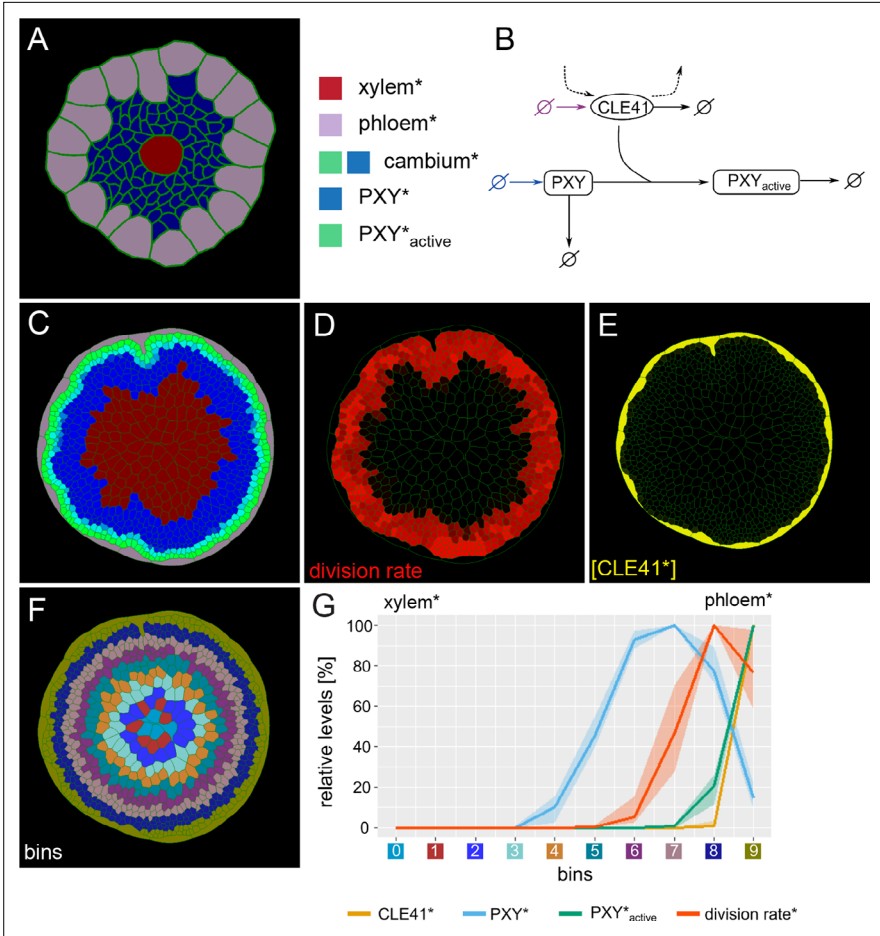

**Figure 1.** Generation of the initial model. (**A**) Tissue template used to run VirtualLeaf simulations. Phloem* is depicted in purple, xylem* in red. Cambium cells* are colored according to their levels of PXY* and PXY-active*. Cambium* is colored in blue due to the initial level of PXY*. Color legend on the right applies to (**A**) and (**C**). (**B**) Schematic representation of the biochemical model. Reactions that occur in all cell types* are drawn in black. Reactions only occurring in the phloem* are depicted in purple, reactions specific to the cambium* are in blue. Crossed circles represent production or degradation of molecules. (**C**) Output of simulation using Model 1. (**D**) Visualization of cell division rates* within the output shown in (**C**).Dividing cells* are marked by red color fading over time. (**E**) Visualization of CLE41* levels within the output shown in (**C**).(**F**) Sorting cells* within the output shown in (C)into bins based on how far their centers are from the center of the hypocotyl*. Different colors represent different bins. (**G**) Visualization of the relative chemical levels and division rates in different bins shown in (F)averaged over n = 10 simulations of Model 1. Each chemical's bin concentration is first expressed as a percentage of the maximum bin value of the chemical and then averaged over all simulations. The colored area indicates the range between minimum and maximum value of the relative chemical concentration. Bin colors along the x-axis correspond to the colors of bins in (**F**).The shading represents the range between minimal and maximal values during simulations.

we took advantage of the observation that the PXY/CLE41 module represses xylem differentiation (*Hirakawa et al., 2008*; *Kondo et al., 2014*). Consequently, we instructed cambium cells* to change their identity into xylem* as soon as they reached a certain size and [PXY$_{active}$*] became lower than a threshold value (*Supplementary file 1*).

In the resulting Model 1, the growing structure maintained a circular pool of dividing cambium cells* with a high concentration of PXY$_{active}$* while producing xylem cells* toward the center of the organ (*Figure 1C*, *Videos 1–4*). As expected, when cambium cells* were displaced to the proximal side of the cambium*, they stopped dividing likely due to low [PXY$_{active}$*] (*Figure 1C and D*, *Video 3*, *Video 4*, *Video 5*) allowing them to reach a size sufficient for xylem* differentiation. Cell* division rates were highest close to CLE41* producing phloem cells* (*Figure 1D–G*, *Video 2*, *Video 3*).

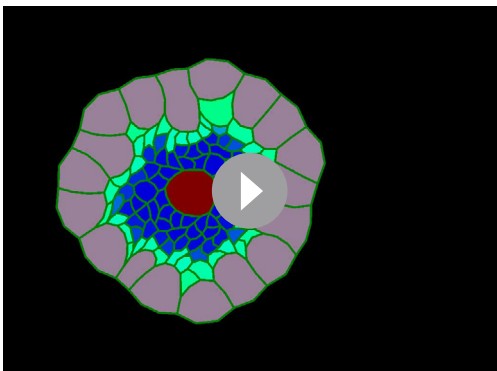

**Video 1.** Model 1 output, visualizing xylem (red) and phloem (purple), and accumulation of PXY* (blue) and PXY$_{active}$* (green).
https://elifesciences.org/articles/66627/figures#video1

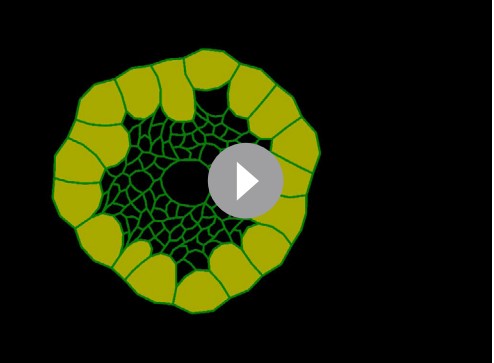

**Video 2.** Model 1 output, visualizing CLE41* (yellow) accumulation.
https://elifesciences.org/articles/66627/figures#video2

Moreover, as PXY$_{active}$* negatively affected the production of PXY*, [PXY*] was particularly low in the distal cambium* region (*Figure 1C, F and G*. *Video 1*, *Video 5*). This pattern was reminiscent of the exclusive activity of the *PXY* promoter in the proximal cambium area observed previously (*Shi et al., 2019*; *Gursanscky et al., 2016*). Thus, although phloem was not produced, with maintaining a circular domain of cambium cells* and cell* proliferation and with promoting xylem* production, Model 1 was able to recapitulate several core features of the active cambium.

## The combination of *PXY* and *SMXL5* promoter reporters reveals cambium anatomy

To identify rules for phloem formation, we took advantage of findings obtained using the *PXYpro:CYAN FLUORESCENT PROTEIN* (*PXYpro:CFP*) and *SUPPRESSOR OF MAX2-LIKE 5pro:YELLOW FLUORESCENT PROTEIN* (*SMXL5pro:YFP*) markers, recently established read-outs for cambium anatomy (*Shi et al., 2019*). *PXYpro:CFP* and *SMXL5pro:YFP* markers label the proximal and distal cambium domain (*Figure 2A*, *Figure 2—figure supplement 1*), respectively, and are therefore indicative of a bipartite cambium organization. *PXYpro:CFP* activity indicates the proximal xylem formation zone whereas *SMXL5pro:YFP* activity indicates the distal phloem formation zone. A narrow central zone in which both markers are active hold cambium stem cells that feed both tissues and also show a high rate of cell divisions in comparison to xylem and phloem progenitors (*Shi et al., 2019*).

To computationally recapitulate the observed maximum of cell division rates in the central cambium domain, we sought to inhibit cell* divisions in the distal layers of the cambium*. Such an effect is, for instance, mediated by the receptor-like kinase MOL1, which, similarly to *SMXL5*, is expressed distally to *PXY* expressing cells and suppresses cambial cell divisions (*Gursanscky et al., 2016*). Because

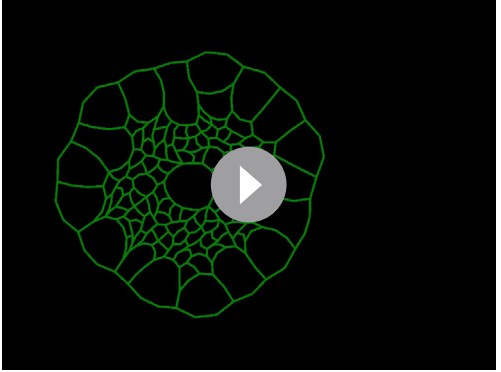

**Video 3.** Model 1 output, visualizing cell divisions (red).
https://elifesciences.org/articles/66627/figures#video3

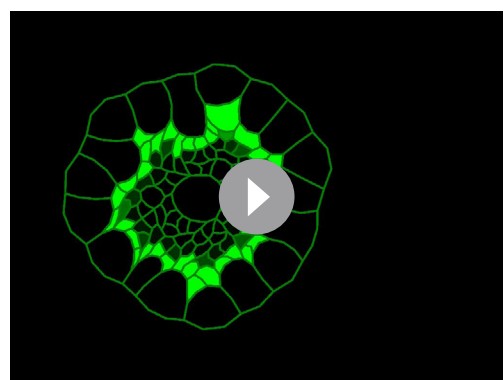

**Video 4.** Model 1 output, visualizing PXY$_{active}$*.
https://elifesciences.org/articles/66627/figures#video4

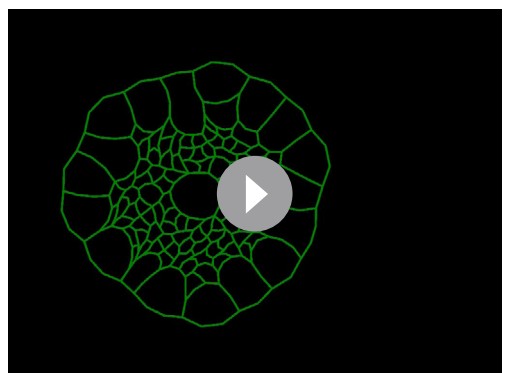

**Video 5.** Model 1 output, visualizing PXY*.
https://elifesciences.org/articles/66627/figures#video5

cells* in the distal cambium* region were characterized by high levels of PXY$_{active}$* (*Figure 1C and G*, *Video 4*), we used PXY$_{active}$* to locally inhibit cell* division and, at the same time, to instruct phloem* formation. Therefore, we modified the rule for cell* differentiation such that, when a cambium cell* reached a specific size, it differentiated into xylem* if [PXY$_{active}$*] became lower than a threshold value and into phloem* if [PXY$_{active}$*] was greater than the same threshold and the cell was larger (*Supplementary file 1*). Thereby, our model followed a classical 'French flag' principle of development according to which concentration gradients of diffusible morphogens pattern surrounding tissues (*Wolpert, 1969*). It is worth noting that the combined effect of CLE41* on cell* proliferation, on phloem* specification, and on [PXY*] may also be achieved by distinct phloem-derived factors mediating these effects individually.

The computational implementation of these rules (Model 2A) resulted in a descending gradient of cell* division rates in the distal cambium* domain likely due to high levels of PXY$_{active}$* (*Figure 2B–E*, *Video 6*, *Video 7*, *Video 8*). The cell* division rate was highest in the central cambium* domain defined by high [PXY*] and by moderate [PXY$_{active}$*] (*Figure 2B–D*, *Video 8*, *Video 9*, *Video 10*, *Video 11*). Also, not only xylem* but also phloem* was continuously produced and the fate of cambium cells* was dependent on their position relative to the differentiated tissues*. In the central cambium* domain, cells* proliferated and constantly replenished the stem cell pool (*Figure 2B*, *Video 6*, *Video 7*, *Video 8*). Thus, by incorporating relatively simple rules, Model 2A comprised major cambium features, including phloem formation. Moreover, in qualitative terms, the resulting anatomy* reproduced the anatomy of a mature *Arabidopsis* hypocotyl (*Figure 2A and E*). It is interesting to note, however, while the cambium domain stays almost perfectly circular in plants, the cambium* in our simulations displayed a clear front instability, suggesting that a stabilizing mechanism exists that we missed in our model.

## Cambium model explains the effect of ectopic CLE41 expression

To evaluate the predictive power of Model 2, we tested its capacity to simulate the effects of genetic perturbation of cambium regulation. Ectopic expression of *CLE41* by employing the *IRREGULAR XYLEM 3/CELLULOSE SYNTHASE CATALYTIC SUBUNIT 7* (*IRX3/CESA7*) promoter, which is active in cells undergoing secondary cell wall deposition (*Mitsuda et al., 2007*; *Taylor et al., 2003*; *Smith et al., 2013*), substantially alters hypocotyl anatomy (*Etchells and Turner, 2010*). This effect was confirmed when *PXYpro:CFP/SMXL5pro:YFP* activities were analyzed in a plant line carrying also an *IRX3pro:CLE41* transgene (*Figure 3A*, *Figure 2—figure supplement 1*, *Figure 3—figure supplement 1*, *Figure 3—figure supplement 2*, *Figure 3—figure supplement 3*, *Figure 3—figure supplement 4*). The *PXYpro:CFP* activity domain had a cylindrical shape including the proximal cambium domain and the xylem tissue itself in plants with a wild-type background (*Figure 2A*, *Figure 3—figure supplement 1*, *Figure 3—figure supplement 2*, *Figure 3—figure supplement 3*, *Figure 3—figure supplement 4*). While in the presence of the *IRX3pro:CLE41* transgene, *PXYpro:CFP* activity was found in irregularly shaped patches containing differentiated xylem vessel elements distributed over the whole cross-section (*Figure 3A*, *Figure 3—figure supplement 1*, *Figure 3—figure supplement 2*, *Figure 3—figure supplement 3*, *Figure 3—figure supplement 4*). Moreover, we observed regions without *PXYpro:CFP* activity in proximal hypocotyl regions where *SMXL5pro:YFP* was active (*Figure 3A*, *Figure 3—figure supplement 1*, *Figure 3—figure supplement 2*, *Figure 3—figure supplement 3*, *Figure 3—figure supplement 4*). Besides, a substantial part of *SMXL5pro:YFP* activity was detected in the distal regions of the hypocotyl forming islands of irregular shape sometimes intermingled with *PXYpro:CFP* activity (*Figure 3A*, *Figure 3—figure supplement 1*, *Figure 3—figure supplement 2*, *Figure 3—figure supplement 3*, *Figure 3—figure supplement 4*). This activity pattern was in contrast to the one found in plants without the *IRX3pro:CLE41* transgene where *SMXL5pro:YFP*

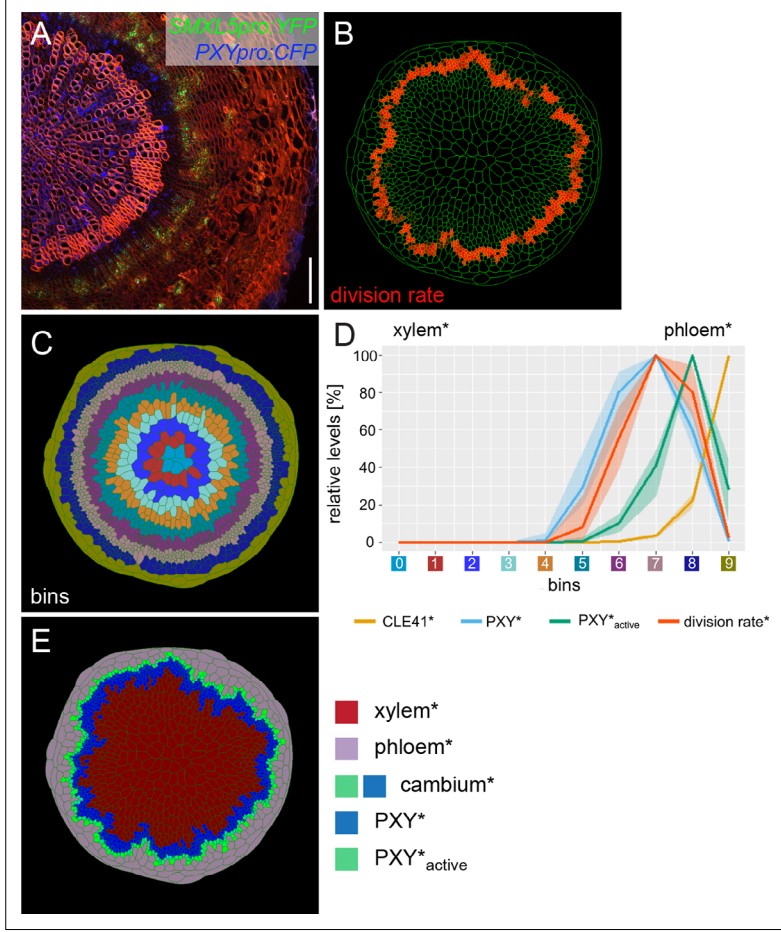

**Figure 2.** Implementing phloem formation into the model. (**A**) Cross-section of a wild-type hypocotyl expressing *PXYpro:CFP* (blue) and *SMXL5pro:YFP* (green). Cell walls are stained by Direct Red 23, mainly visualizing xylem (red). Only a sector of the hypocotyl is shown with the center on the left. Scale bar: 100µm. Ten samples were analysed in total with similar results. An image version for color-blind readers is provided in ***Figure 2—figure supplement 1***. (**B**) Visualization of cell division rates* within the output shown in (**C, E**).Dividing cells* are marked by red color fading over time. (**C**) Sorting cells* within the output shown in (B, E)into bins. (**D**) Visualization of the average relative chemical levels* and division rates* in different bins of repeated simulations of Model 2A (n = 10). Bin label colors along the x-axis correspond to the colors of bins shown in (**C**).The shading represents the range between minimal and maximal values during simulations. (**E**) Output of simulation using Model 2A. Unlike Model 1 (***Figure 1C***), Model 2A produces new phloem cells*.

The online version of this article includes the following source data and figure supplement(s) for figure 2:

**Figure supplement 1.** Wild type control for fluorescent reporter analyses and color-blind modes for images shown in main figures.

**Figure supplement 1—source data 1.** Raw image data collected for ***Figure 2—figure supplement 1***.

reporter activity surrounded the *PXYpro:CFP* expression domain only from the distal side (***Figure 2A***, ***Figure 3—figure supplement 1***, ***Figure 3—figure supplement 2***, ***Figure 3—figure supplement 3***, ***Figure 3—figure supplement 4***). These results indicated that not only the radial symmetry of the hypocotyl (***Etchells and Turner, 2010***) but also cambium organization depends on the site of CLE41 production.

To simulate the effect of the *IRX3pro:CLE41* transgene in silico, we instructed xylem cells* to produce CLE41* at the same rate as phloem cells* (Model 2B). Although in this case xylem* formation was initially repressed possibly due to high levels of $PXY_{active}$* in all cambium* cells (***Figure 3B***, ***Video 12***, ***Video 13***, ***Video 14***), new xylem cells* were formed as soon as the distance between existing xylem and phloem cells* became large enough such that CLE41* levels and, in turn, [$PXY_{active}$*] dropped to

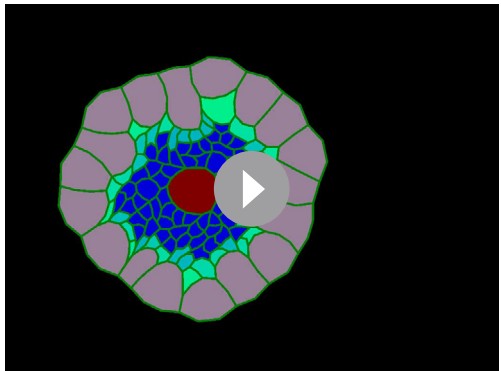

**Video 6.** Model 2A output, visualizing xylem (red) and phloem (purple), and accumulation of PXY* (blue) and PXY-active* (green).
https://elifesciences.org/articles/66627/figures#video6

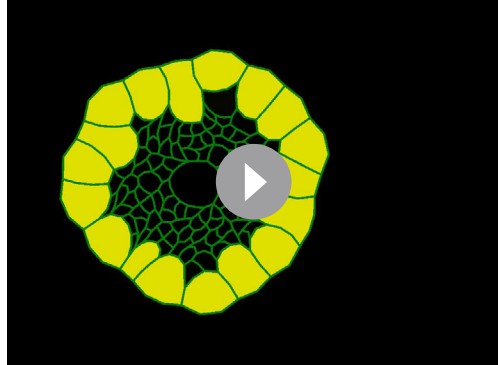

**Video 7.** Model 2A output, visualizing CLE41* (yellow) accumulation.
https://elifesciences.org/articles/66627/figures#video7

permissive levels (*Figure 3C*, *Video 12*, *Video 13*, *Video 14*). New phloem cells* were produced close to existing phloem and xylem cells* likely due to high levels of PXY_active* (*Figure 3C*, *Video 15*, *Video 16*). As a result, Model 2B produced a similar disruption in cambium* organization, as observed in *IRX3pro:CLE41* plants (*Figure 3D*, *Video 12*, *Video 13*, *Video 14*). Zones with both high [PXY_active*] and low [PXY*], which were found in the distal cambium* in Model 2A (*Figure 2E*, *Video 16*, *Video 17*), appeared in the organ* center together with individual xylem cells* (*Figure 3D*). Moreover, in addition to being produced in distal regions, new phloem cells* were produced in the central areas of the organ* as demonstrated previously for *IRX3pro:CLE41* plants (*Etchells and Turner, 2010*). Thus, rules determining cambium* polarity implemented in Model 2 were sufficient to simulate organ anatomy found in wild-type and *IRX3pro:CLE41* genetic backgrounds.

In contrast, a discrepancy between the model logic and the in planta situation was suggested when we compared a model having reduced PXY* activity with *pxy* mutants carrying the *PXYpro:CFP* and *SMXL5pro:YFP* reporters. In *pxy* mutants, the xylem tissue did not have a cylindrical shape, but was instead clustered in radial sectors showing *PXYpro:CFP* and, at their distal ends, *SMXL5pro:YFP* activity, whereas regions in between those sectors had little to no xylem and did not show reporter activity (*Figure 3E*, *Figure 2—figure supplement 1*, *Figure 3—figure supplement 1*, *Figure 3—figure supplement 2*, *Figure 3—figure supplement 3*, *Figure 3—figure supplement 4*). Interestingly, *PXYpro:CFP* and *SMXL5pro:YFP* activity domains were still mostly distinct meaning that *PXYpro:CFP* activity did not expand further beyond established xylem than in wild type (*Figure 3E*, *Figure 3—figure supplement 5*). This discrepancy indicated that, in contrast to our assumption, the CLE41-PXY signaling module did not restrict *PXY* promoter activity in the distal cambium. Of note, the sharp border between *PXYpro:CFP* and

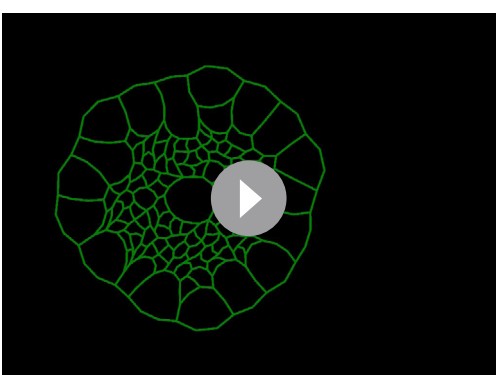

**Video 8.** Model 2A output, visualizing cell divisions (red).
https://elifesciences.org/articles/66627/figures#video8

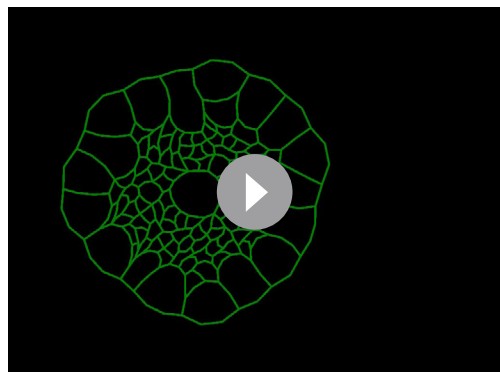

**Video 9.** Model 2A output, visualizing cell divisions (red) together with PXY* (blue) and PXY-active* (green) accumulation.
https://elifesciences.org/articles/66627/figures#video9

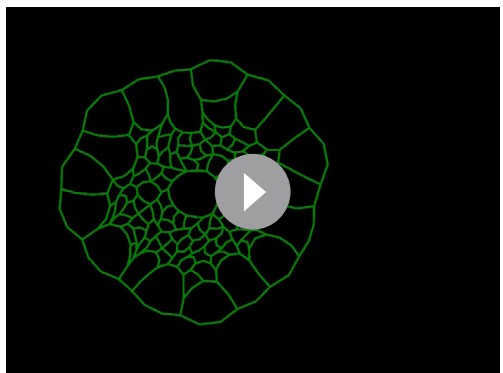

**Video 10.** Model 2A output, visualizing PXY_active*.
https://elifesciences.org/articles/66627/figures#video10

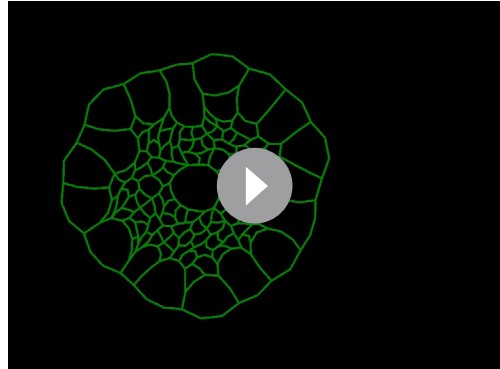

**Video 11.** Model 2A output, visualizing PXY*.
https://elifesciences.org/articles/66627/figures#video11

*SMXL5pro:YFP* activity was less pronounced in *pxy* mutants mostly due to a spread of *SMXL-5pro:YFP* activity towards xylem tissues (*Figure 3—figure supplement 5*). The discrepancy between Model 2 and the situation in plants was confirmed when we completely eliminated PXY* activity from our model (Model 2C). As expected, this elimination resulted in the absence of growth due to the full dependence of cell* divisions on the PXY* function, clearly being at odds with the phenotype of *pxy* mutants (*Figure 3E*). Even when we only reduced PXY* activity (Model 2D), this did not result in a split of the continuous cambium domain* but abolished phloem* formation and increased the production of xylem* (*Figure 3F*).

Interestingly, the quantification of water transporting xylem vessels, xylem fibers, which provide mechanical stability, and xylem parenchyma in sections from wild-type and *pxy* mutant hypocotyls by automated image segmentation revealed that the total number of xylem cells and the number of xylem vessels was comparable (*Figure 3G–I*, *Figure 3—figure supplement 6*). In contrast, the number of cells classified as fibers was substantially reduced in *pxy* mutants, whereas the number of cells classified as parenchyma was increased (*Figure 3G–I*). These results suggested that during radial growth *PXY* promotes the fiber-parenchyma ratio in the xylem, while the formation of xylem vessels and the total number of cambium-derived cells produced toward the xylem is hardly *PXY*-dependent.

## Multiple phloem-derived factors determine cambium activity

Our observations prompted us to reconsider some features of the model and to extend our 'French flag' approach. As the proximal cell production rate by the cambium was not *PXY*-dependent, we made xylem* formation independent from the control of PXY-active*. Instead, cambium cells* differentiated into xylem cells* when they reached a specific size and, at the same time, expressed PXY* as a positional feature. To maintain a population of active cambium cells* in the absence of PXY*, we introduced a second phloem*-derived factor (PF), reminiscent of the PEAR transcription factors identified recently (*Miyashima et al., 2019*). PF* stimulated cell* divisions by promoting the production of a division factor (DF) in cambium cells* and in phloem parenchyma* (*Figure 4A*, *Figure 4—figure supplement 1*, see below). Cambium cells* divided only if the concentration of DF* exceeded a threshold value (*Supplementary file 1*). DF* production was at the same time stimulated by PXY_active* as its only effect in cambium cells* (*Figure 4A*). Thereby, cambial cell* divisions were dependent on the combined influence of PXY_active* and their proximity to phloem poles* (see below).

PF* was, thus, produced in phloem poles* and the levels in other cells* were determined by the diffusion and degradation:

$$\frac{d}{dt}\left[PF^*\right] = production_{PF} + diffusion_{PF} - degradation_{PF} \cdot \left[PF^*\right] \qquad (5)$$

DF* production was, in turn, determined as follows:

$$\frac{d}{dt}\left[DF^*\right] = diffusion_{DF} + production_{DF} \qquad (6)$$

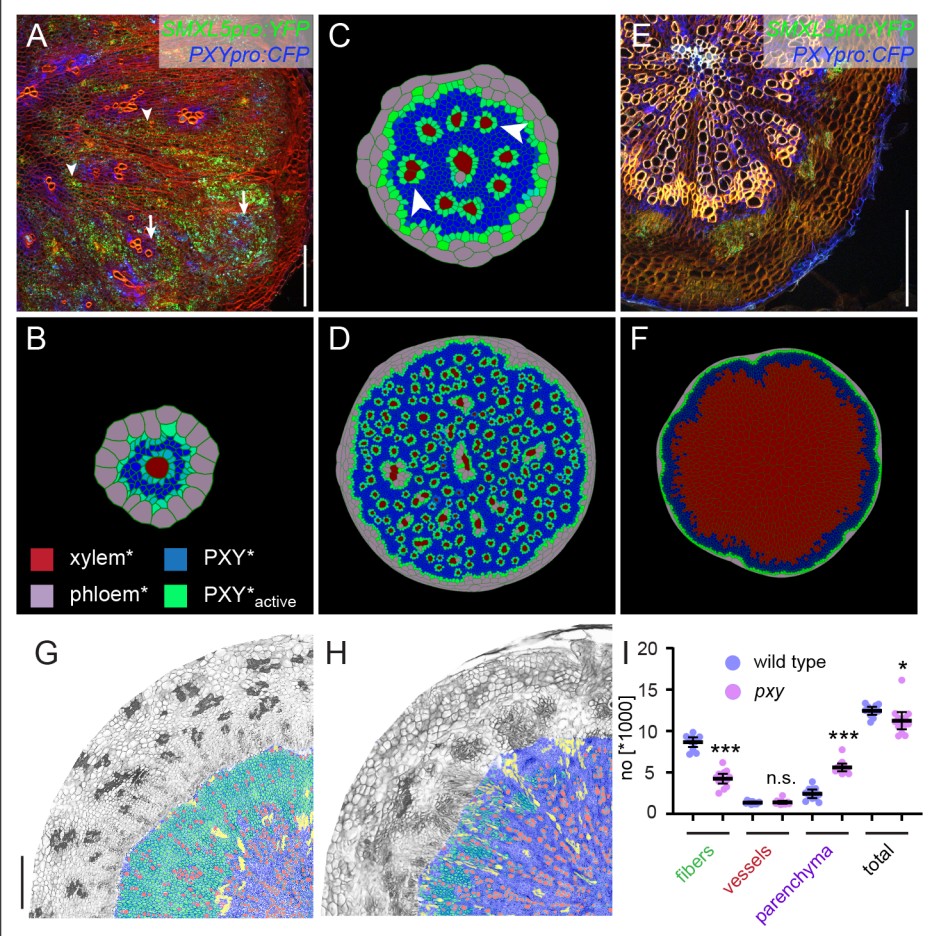

**Figure 3.** Comparing the effect of perturbing cambium activity in the model and in plants. (**A**) Cross-section of a hypocotyl carrying *PXYpro:CFP* (blue), *SMXL5pro:YFP* (green) markers, and the *IRX3pro:CLE41* transgene. Cell walls are stained by Direct Red 23 visualizing mostly xylem (red). Arrowheads point to proximal hypocotyl regions where *SMXL5pro:YFP* activity is found. Arrows indicate distal regions with *SMXL5pro:YFP* activity. Cell walls are stained by Direct Red 23 visualizing mostly xylem (red). Only a quarter of the hypocotyl is shown with the center in the upper-left corner. Scale bar: 100µm. Ten samples were analysed in total with similar results. An image version for color-blind readers is provided in *Figure 2—figure supplement 1*. (**B**) First frames of Model 2B simulations. Due to the expression of CLE41* by xylem cells*, high levels of PXY-active* are generated around xylem cells* already at this early stage. Legend in (**B**) indicates color code in (**B, C, D, F**). (**C**) Intermediate frames of Model 2B simulations. Newly formed xylem cells* express CLE41* and produce high levels of PXY-active* next to them (white arrowheads). (**D**) The final result of Model 2B simulations. Zones of PXY* (blue) and PXY-active* (green) are intermixed, xylem cells* are scattered, and phloem cells* are present in proximal areas of the hypocotyl*. (**E**) Cross-section of a *pxy* mutant hypocotyl carrying *PXYpro:CFP* (blue) and *SMXL5pro:YFP* (green) markers, stained by Direct Red 23 (red). The xylem shows a ray-like structure. Only a quarter of the hypocotyl is shown with the center in the upper-left corner. Ten samples were analysed in total with similar results. Scale bar: 100µm. An image version for color-blind readers is provided in *Figure 2—figure supplement 1*. (**F**) Final result of Model 2D simulations. Reducing PXY* levels leads to similar results as produced by Model 1 (*Figure 1C*) where only xylem* is produced. (**G, H**) Comparison of histological cross-sections of a wild-type (G)and a *pxy* (H)mutant hypocotyl, including cell-type classification produced by ilastik. The ilastik classifier module was trained to identify xylem vessels (red), fibers (green), and parenchyma (purple), unclassified objects are shown in yellow. Size bar in G: 100 µm. Same magnification in G and H. (**I**) Comparison of the number of xylem vessels, fibers, and parenchyma cells found in wild-type (blue) and *pxy* mutants (purple). Welch's *t*-test was performed comparing wild-type and *pxy* mutants for the different cell types (n = 11–12). ***p<0.0001, *p<0.05. Lines indicate means with a 95% confidence interval. 11 (wild type) and 13 (*pxy*) samples were analysed.

The online version of this article includes the following source data and figure supplement(s) for figure 3:

**Source data 1.** Source data for cell type classification using ilastik.

*Figure 3 continued on next page*

*Figure 3 continued*

**Figure supplement 1.** First example revealing the dynamics of *PXYpro:CFP/SMXL5pro:YFP* activities during radial hypocotyl growth in wild-type, *IRX3pro:CLE41,* and *pxy* plants.

**Figure supplement 1—source data 1.** Raw image data collected for *Figure 3—figure supplement 1*, first part.

**Figure supplement 1—source data 2.** Raw image data collected for *Figure 3—figure supplement 1*, second part.

**Figure supplement 2.** Second example revealing the dynamics of *PXYpro:CFP/SMXL5pro:YFP* activities during radial hypocotyl growth in wild-type, *IRX3pro:CLE41,* and *pxy* plants.

**Figure supplement 2—source data 1.** Raw image data collected for *Figure 3—figure supplement 2*, first part.

**Figure supplement 2—source data 2.** Raw image data collected for *Figure 3—figure supplement 2*, second part.

**Figure supplement 3.** First example revealing the dynamics of *PXYpro:CFP/SMXL5pro:YFP* activities during radial hypocotyl growth in wild-type, *IRX3pro:CLE41,* and *pxy* plants (color-blind mode).

**Figure supplement 4.** Second example revealing the dynamics of *PXYpro:CFP/SMXL5pro:YFP* activities during radial hypocotyl growth in wild-type, *IRX3pro:CLE41,* and *pxy* plants (color-blind mode).

**Figure supplement 5.** Close-up revealing the dynamics of *PXYpro:CFP/SMXL5pro:YFP* activities in hypocotyls in wild-type and *pxy* plants.

**Figure supplement 6.** Overview and magnifications of sections used for cell-type classification shown in *Figure 3*.

where K stands for an empirically defined parameter capping the production rate of DF*.

Based on the strong association of xylem sectors with developing phloem cells (*Figure 3E*), we further hypothesized that the formation of those sectors in *pxy* mutants was dependent on the heterogeneity of cell-type distribution in the phloem. Therefore, phloem cells* from the previous models were split into two cell types – phloem parenchyma* and phloem poles* (*Figure 4A*, *Figure 4—figure supplement 1*). To achieve the dispersed pattern of phloem poles, cambium-derived cells* fulfilling the criteria to differentiate into phloem* (see above), differentiated into phloem poles* by default, unless inhibited by PF*, which was specifically produced in pole cells* (*Supplementary file 1*). Thereby, phloem poles* suppressed phloem pole* formation in their vicinity, expected to result in a patchy pattern of phloem poles as observed in planta (*Sankar et al., 2014*). The inhibition of phloem poles in their immediate environment is reminiscent to the CLE45/RECEPTOR LIKE PROTEIN KINASE 2 (RPK2) signaling cascade restricting protophloem sieve element identity to its usual position (*Gujas et al., 2020*; *Qian et al., 2022*). It is worth noting that in our model CLE41* was still produced in both phloem poles* and phloem parenchyma* but with a higher rate in phloem poles*. To further achieve PXY*-independent cambium subdomain separation, phloem parenchyma* and phloem poles* were set to express another diffusive signal (RP) that suppressed PXY* expression in cambium* cells, the role that was played by PXY$_{active}$* before (*Figure 4A*, *Figure 4—figure supplement 1*, *Supplementary file 1*). The role of RP is reminiscent to the role of cytokinin that inhibits xylem-related features in tissue domains designated for phloem development

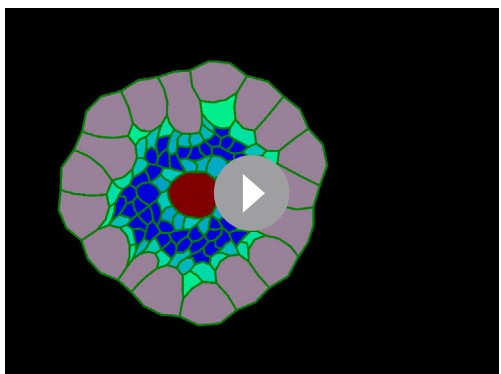

**Video 12.** Model 2B output, visualizing xylem (red) and phloem (purple), and accumulation of PXY* (blue), and PXY-active* (green).

https://elifesciences.org/articles/66627/figures#video12

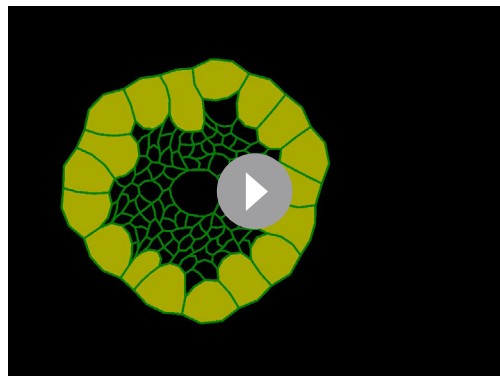

**Video 13.** Model 2B output, visualizing CLE41* (yellow) accumulation.

https://elifesciences.org/articles/66627/figures#video13

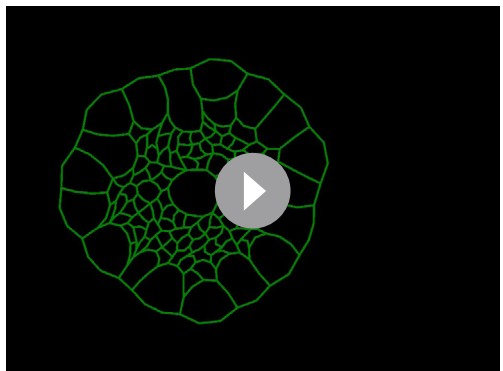

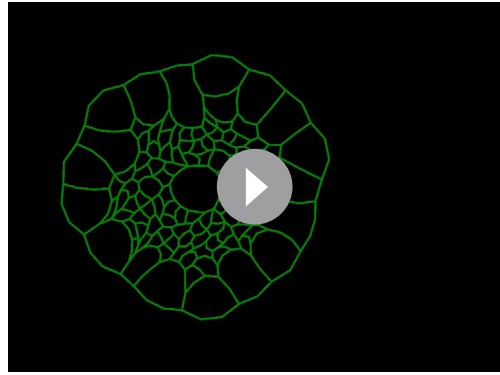

**Video 14.** Model 2B output, visualizing cell divisions (red).
https://elifesciences.org/articles/66627/figures#video14

**Video 15.** Model 2B output, visualizing accumulation of PXY* (blue) and PXY-active* (green).
https://elifesciences.org/articles/66627/figures#video15

(*Mähönen et al., 2006*). Importantly, cell divisions in the distal cambium* were not actively repressed anymore but were exclusively dependent on cell size and the level of DF* (*Supplementary file 1*).

The implementation of these principles in silico (Model 3A) resulted again in the establishment of two cambium* subdomains – the distal subdomain that was characterized by high concentrations of DF* and the proximal subdomain characterized by high PXY* concentration (*Figure 4B–D*, *Figure 4— figure supplement 1*, *Video 18*, *Video 19*, *Video 20*, *Video 21*, *Video 22*, *Video 23*). Distally, the cambium* produced phloem parenchyma cells* from which phloem poles* were continuously formed with a pattern resembling the patchy phloem pattern observed in plants (*Figure 4B*, *Figure 4— figure supplement 1*; *Sankar et al., 2014*; *Wallner et al., 2020*). Interestingly, the localization of PF* production mainly in phloem poles* resulted in increased DF levels in the vicinity of those poles and, consequently, in locally increased cell* division rates (*Video 20*, *Video 21*). This observation is in line with the observation that phloem poles drive cell divisions in their immediate environment and that phloem cells still divide after initial specification (*Miyashima et al., 2019*; *Bossinger and Spokevicius, 2018*). When comparing the radial pattern of *PXYpro:CFP/SMXL5pro:YFP* activities and, as an in silico approximation to these activities, the distribution of PXY* and DF* in our model over time, patterns were stable in both cases (*Figure 4B*, *Figure 3—figure supplement 1*, *Figure 3—figure supplement 2*, *Figure 3—figure supplement 3*, *Figure 3—figure supplement 4*, *Video 18*, *Video 22*, *Video 23*). This demonstrated that our model was able to generate stable radial patterns of gene* activity comparable to the in planta situation.

By instructing CLE41* production additionally in xylem cells*, we next simulated CLE41-misexpression by the *IRX3* promoter (Model 3B, *Figure 4—figure supplement 1*, *Figure 4E*, *Video 24*, *Video 25*, *Video 26*, *Video 27*, *Video 28*, *Video 29*). CLE41* interacted with PXY* on the proximal cambium* border, which resulted in ectopic DF* production and phloem-parenchyma* formation in the proximal hypocotyl* regions (*Figure 4E*, *Video 24*, *Video 28*), similarly as during

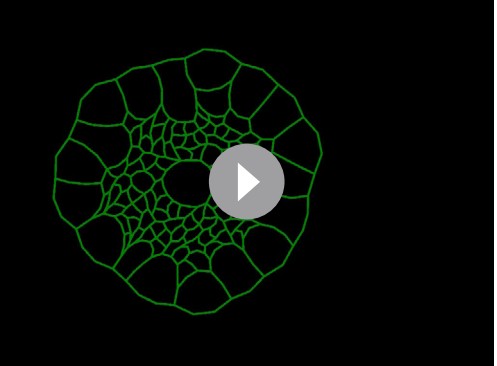

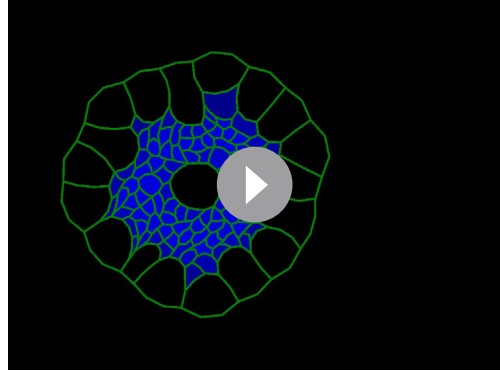

**Video 16.** Model 2B output, visualizing PXY$_{active}$*.
https://elifesciences.org/articles/66627/figures#video16

**Video 17.** Model 2B output, visualizing PXY*.
https://elifesciences.org/articles/66627/figures#video17

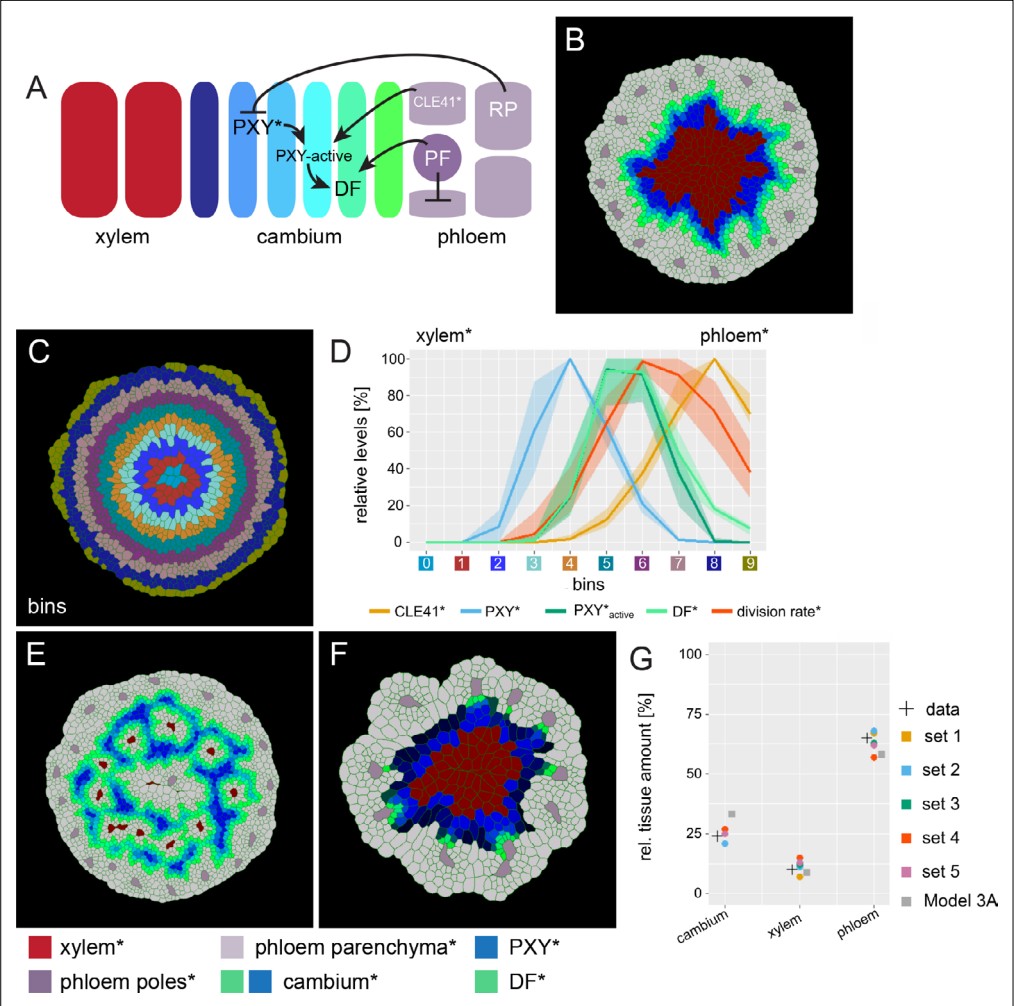

**Figure 4.** An extended model for simulating genetic perturbations of cambium activity. (**A**) Regulatory network proposed based on experimental observations. (**B**) Result of the simulation run for Model 3A. This model implements the network interactions described in (**A**). Color coding at the bottom of Figure 4. (**C**) Outline of cell bins for the results of Model 3A, as shown in (**B**).(**D**) Visualization of the relative levels of chemicals* and division rates* in different bins. Bin colors along the x-axis correspond to the bin colors in (**C**).The shading represents the range between minimal and maximal values during simulations (n = 10). (**E**) Output of Model 3B simulation. Ectopic CLE41* expression was achieved by letting xylem cells* produce CLE41*. (**F**) Output of Model 3C. Simulation of the *pxy* mutant was achieved by removing the stimulation of DF* production by PXY* and hence by removing the effect of PXY* on cell division and cambium* subdomain patterning. Because of the network structure, PXY* can be eliminated from Model 3 without letting the model collapse (*Figure 3F*) but reproducing the *pxy* mutant phenotype observed in adult hypocotyls (Figure 4E). Be aware that cell* proliferation is generally impaired under these conditions reducing overall template growth*. Because the final output covers the same image area, cell size seems to be enlarged which, however, is not the case. (**G**) Estimated tissue ratios for five identified parameter sets compared to experimental values ('data') found for wild type hypocotyls 20d after germination (*Sankar et al., 2014*) and compared to the final model output before the automated parameter search ('Model 3A') and the implementation of experimentally determined cell wall thickness for xylem* and phloem*.

The online version of this article includes the following figure supplement(s) for figure 4:

**Figure supplement 1.** Overview of cell types*, regulatory interactions and expression* profiles in Model 3.

**Figure supplement 2.** Determination of cell wall thickness across the radial sequence of hypocotyl tissues.

**Figure supplement 3.** Behavior of the different model parameterizations (Model 4:2–5).

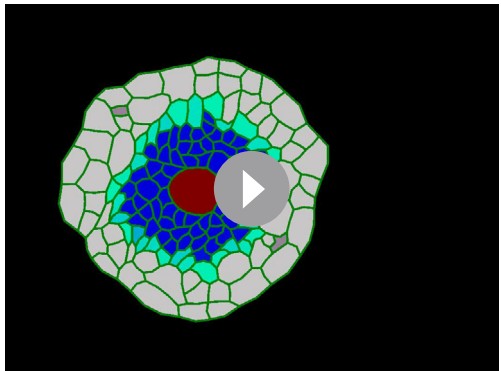

**Video 18.** Model 3A output, visualizing xylem (red), phloem parenchyma (light purple), and phloem poles (dark purple), and accumulation of PXY* (blue) and the division chemical (DF)* (green).

https://elifesciences.org/articles/66627/figures#video18

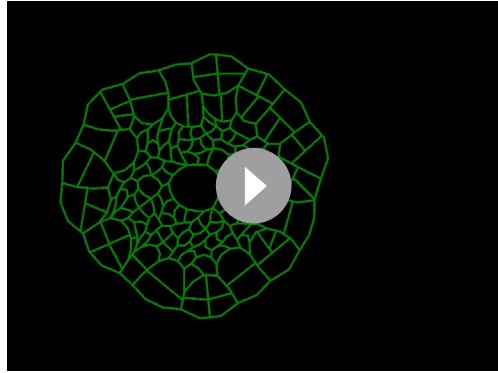

**Video 19.** Model 3A output, visualizing CLE41* (yellow) accumulation.

https://elifesciences.org/articles/66627/figures#video19

radial hypocotyl growth in *IRX3pro:CLE41* plants (*Figure 3A*, *Figure 3—figure supplement 1*, *Figure 3—figure supplement 2*, *Figure 3—figure supplement 3*, *Figure 3—figure supplement 4*). Still, xylem cells* were formed, generating a patchy xylem* pattern resembling the xylem configuration found in *IRX3pro:CLE41* plants (*Figure 3A*, *Figure 4E*, *Video 24*).

Fully eliminating CLE41* binding to PXY* and therefore PXY* activity but keeping the positional information of PXY* for xylem cell differentiation (Model 3C, *Figure 4—figure supplement 1*) generated a patchy outline of the distal cambium* subdomain (*Figure 4F*, *Video 30*, *Video 31*, *Video 32*, *Video 33*, *Video 34*, *Video 35*). While PXY* was usually the main trigger of cell* divisions in cambium cells* at a certain distance from phloem poles*, PF* was sufficient for triggering cell divisions next to phloem poles*. Heterogeneous cambium activity was already observable at early phases of radial hypocotyl growth in silico and in planta and resulted overall in a reduced tissue production in both systems (*Figure 4F*, *Figure 3—figure supplement 1*, *Figure 3—figure supplement 2*, *Figure 3—figure supplement 3*, *Figure 3—figure supplement 4*, *Video 30*, *Video 31*, *Video 32*, *Video 33*, *Video 34*, *Video 35*). Thus, by introducing both a PXY*-independent pathway stimulating cambium* proliferation and a dependence of cell* proliferation on the distance to phloem poles*, we were able to simulate important features of the *pxy* mutant phenotype (*Figure 3E*, *Figure 4F*, *Figure 3—figure supplement 1*, *Figure 3—figure supplement 2*, *Figure 3—figure supplement 3*, *Figure 3—figure supplement 4*). Collectively, we concluded that we established a computational cambium model sufficiently robust to simulate major genetic perturbations of cambium regulation.

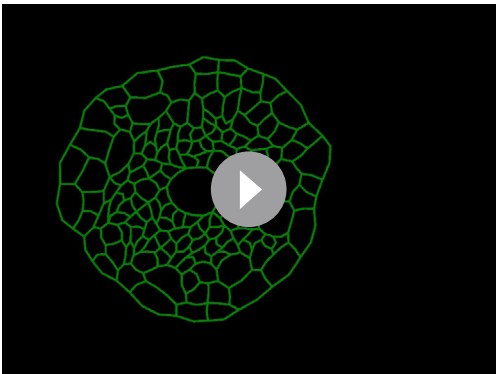

**Video 20.** Model 3A output, visualizing cell divisions (red).

https://elifesciences.org/articles/66627/figures#video20

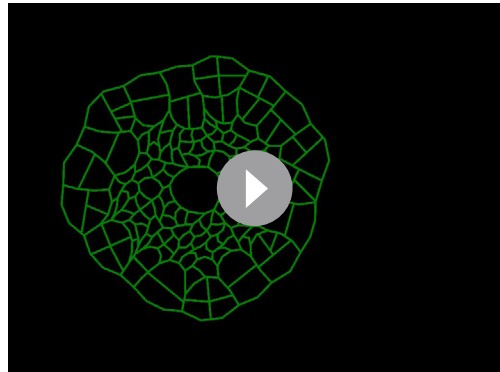

**Video 21.** Model 3A output, visualizing accumulation of PXY* (blue) and PXY$_{active}$* (green).

https://elifesciences.org/articles/66627/figures#video21

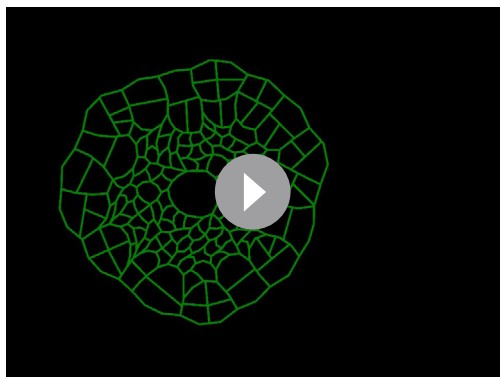

**Video 22.** Model 3A output, visualizing PXY<sub>active</sub>*.
https://elifesciences.org/articles/66627/figures#video22

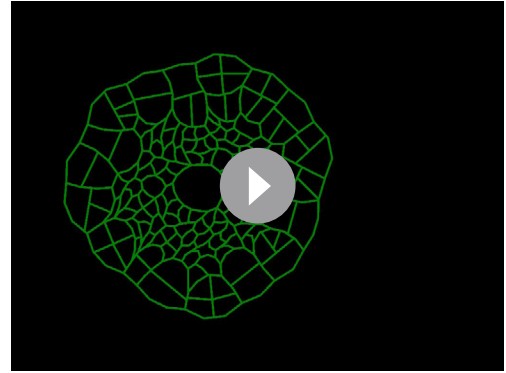

**Video 23.** Model 3A output, visualizing PXY*.
https://elifesciences.org/articles/66627/figures#video23

## Physical properties of cambium-derived cells have the potential to influence stem cell behavior

Next, we were interested to see whether the established model was able to reveal organ-wide features of radial plant growth. A characteristic of cambium stem cells is that they divide mostly in periclinal orientation, which is in parallel to the organ surface, resulting in the frequent formation of radial cell files (*Figure 2A*). Interestingly, although the overall tissue anatomy of the modeled organ* resembled the in planta situation, cell division orientation in our model outputs was almost random, suggesting that radial cell file formation cannot be explained by the molecular signaling pathways implemented into the model (*Figure 4A*). The strong dominance of periclinal divisions in planta, however, implies the presence of a positional signal instructing cell division orientation. Because classical observations indicated that physical forces play a role in this regard (*Brown, 1964*; *Brown and Sax, 1962*; *Lintilhac and Vesecky, 1984*), we tested whether the model was suited for finding indications for the influence of differential cell stiffness on geometric features of radial plant growth.

To do so, we first determined the relative cell wall thickness in hypocotyl cross-sections using the cell wall dyes Direct Yellow 96 and Direct Red 23 (*Ursache et al., 2018*) as an indication. Notably, staining intensities were approximately half as strong compared to cells of the surrounding tissue (*Figure 4—figure supplement 2*). Expecting that staining intensities correlate with cell wall stiffness and by also taking into account results obtained previously by atomic force microscopy of the cambium region (*Arnould et al., 2022*), we assumed that cambium stem cells are half as stiff as surrounding cells and integrated this feature into our model by expanding VirtualLeaf to allow for the integration of cell-type specific wall stiffness (see Supporting Information 'VirtualLeaf Simulations' for details). We implemented this information in the Hamiltonian operator, which is used to approximate the

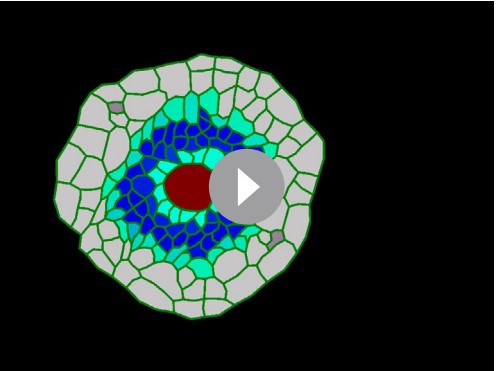

**Video 24.** Model 3B output, visualizing xylem (red), phloem parenchyma (light purple), and phloem poles (dark purple), and accumulation of PXY* (blue) and the division chemical (DF)* (green).
https://elifesciences.org/articles/66627/figures#video24

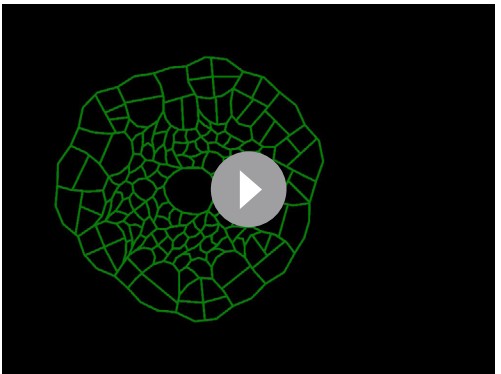

**Video 25.** Model 3B output, visualizing CLE41* (yellow) accumulation.
https://elifesciences.org/articles/66627/figures#video25

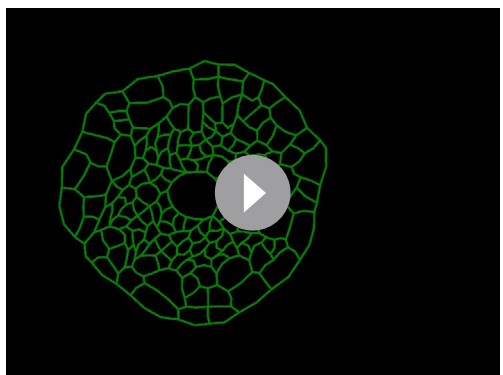

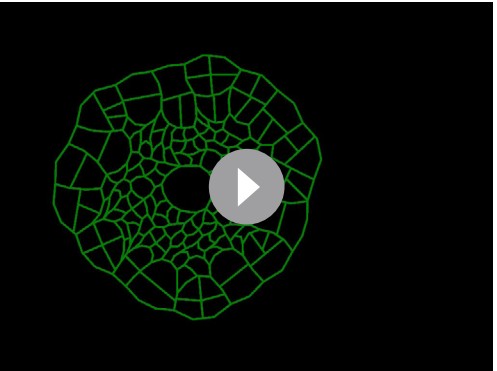

**Video 26.** Model 3B output, visualizing cell divisions (red).
https://elifesciences.org/articles/66627/figures#video26

**Video 27.** Model 3B output, visualizing accumulation of PXY* (blue) and PXY$_{active}$* (green).
https://elifesciences.org/articles/66627/figures#video27

energy of the system and takes both turgor pressure and cell wall resistance into account. In practice, this means that a higher cell wall stiffness will increase the cell walls' resistance to being stretched and will result in slower cell* growth.

Utilizing this expanded model (Model 4), we investigated the parameter space to find parameters accurately describing cambium activity not only qualitatively but also quantitatively. To incorporate realistic tissue ratios and unbiased parameter identification, we performed an automated parameter search using a previous characterization of *Arabidopsis* hypocotyl anatomy (*Sankar et al., 2014*) as a criterion for parameter selection. To this end, we evaluated our searched parameter sets to aim for a cell-type distribution of 24, 10, and 65% for cambium*, xylem*, and phloem cell* number, respectively. Performing 12,500 simulations resulted in n = 5 parameter sets (*Supplementary file 2*), which produced more realistic cell-type proportions than we achieved by our manually selected set before (*Figure 4G*, *Figure 4—figure supplement 3*). Thus, by taking real cell-type proportions as a guideline for parameter search, we were able to establish a model generating a more realistic morphology as a solution. Furthermore, by generating several parameter sets that described the experimentally observed tissue ratios equally well, we demonstrated that even with differing parameter values the model behavior remained consistent reaffirming the model structure we had identified with Model 3A and was parameterized in Model 4 (*Figure 5A and B*, *Figure 4—figure supplement 3*, *Video 36*, *Video 37*, *Video 38*, *Video 39*, *Video 40*, *Video 41*).

To next investigate the role of biomechanics in the direction of cell division, we analyzed the model behavior at different cell wall stiffness values. Specifically, we were interested in the role of xylem* and epidermis*, the latter being represented by the relative perimeter stiffness of the outer tissue boundary in VirtualLeaf. Of note, defining the outer cell* layer as epidermis* was done for simplicity reasons as the rather complex periderm usually forms the outer tissues of older hypocotyls (*Serra et al., 2022*). Here, we assumed xylem or epidermis cells* and, in turn, the relative perimeter stiffness to be more

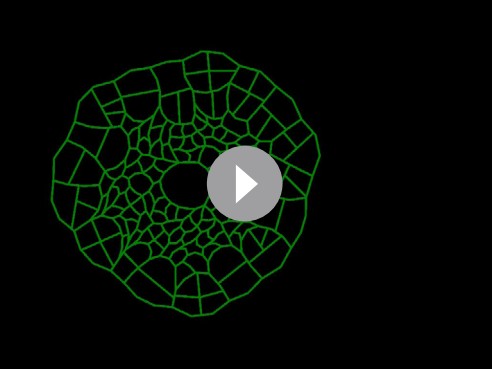

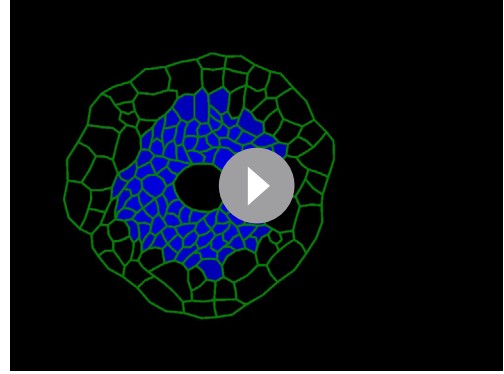

**Video 28.** Model 3B output, visualizing PXY$_{active}$*.
https://elifesciences.org/articles/66627/figures#video28

**Video 29.** Model 3B output, visualizing PXY*.
https://elifesciences.org/articles/66627/figures#video29

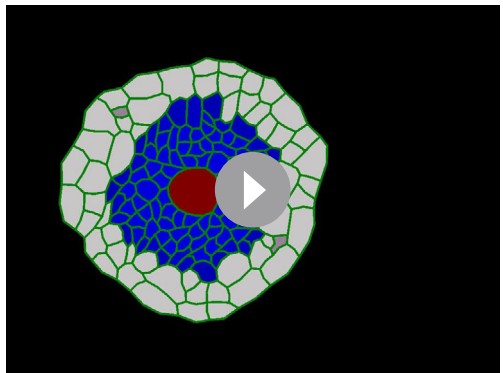

**Video 30.** Model 3C output, visualizing xylem (red), phloem parenchyma (light purple), and phloem poles (dark purple), and accumulation of PXY* (blue) and the division chemical (DF)* (green).
https://elifesciences.org/articles/66627/figures#video30

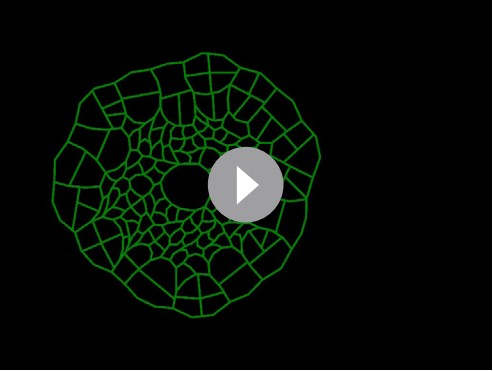

**Video 31.** Model 3C output, visualizing CLE41* (yellow) accumulation.
https://elifesciences.org/articles/66627/figures#video31

resistant to expansion due to the thickness of their cell walls* and implemented this behavior in Model 4 as a cellular* property. First we explored how the variation of the stiffness of xylem cell walls* impacted tissue formation (*Figure 5C*, *Video 42*, *Video 43*, *Video 44*, *Video 45*, *Video 46*, *Video 47*), we first observed that, as expected, increasing cell wall* stiffness led to a xylem*-specific decrease in cell* size and major axis length (*Figure 5—figure supplement 1A and B*). In turn, some cambium cells showed an increase in length as the cell type* with the closest proximity to xylem cells*. In addition, we observed a general decrease in the number of cells*, particularly of xylem cells* (*Figure 5—figure supplement 1D*). We explained this effect by a 'physical' constraint generated by 'stiffer' xylem cells* acting on neighboring cambium cells* impairing their expansion and, thus, their transformation into xylem* (*Video 42*, *Video 43*, *Video 44*, *Video 45*, *Video 46*, *Video 47*). Importantly, neither cell area nor cell length was affected in phloem cells* and the number of cambium cells* stayed constant (*Figure 5—figure supplement 1*), suggesting that the general growth dynamics of the model and especially the behavior of cambium cells* was comparable under the different stiffness* regimes. When analyzing the same characteristics for the different epidermis* tissue regimes (*Figure 5D*, *Video 48*, *Video 49*, *Video 50*, *Video 51*, *Video 52*), we found that neither cell size* nor cell length* were impacted (*Figure 5—figure supplement 2*). Instead, we found a decrease in the number of cells* per simulation with increasing cell wall stiffness, in particular phloem parenchyma* and phloem pole cells*, as the increased resistance of the outer tissue boundary limited the overall tissue growth, resulting in less cells being produced in the outer parts of the tissue (*Figure 5—figure supplement 2*).

To access the effect of increased stiffness of xylem and epidermis cell walls* on cell* division orientation, we first defined cell lineages* as groups of cells* having originated from the same precursor

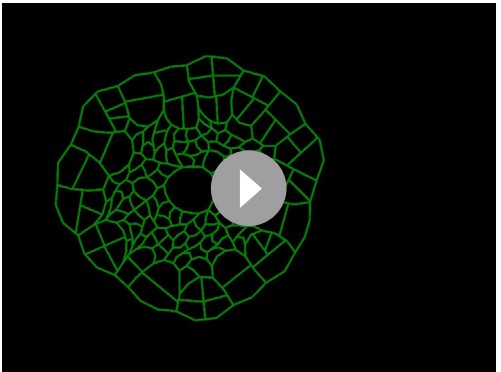

**Video 32.** Model 3C output, visualizing cell divisions (red).
https://elifesciences.org/articles/66627/figures#video32

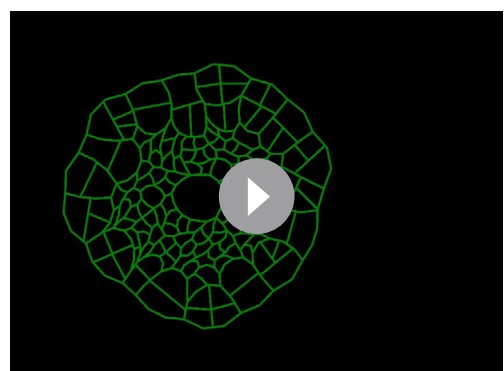

**Video 33.** Model 3C output, visualizing accumulation of PXY* (blue) and the division chemical (DF)* (green).
https://elifesciences.org/articles/66627/figures#video33

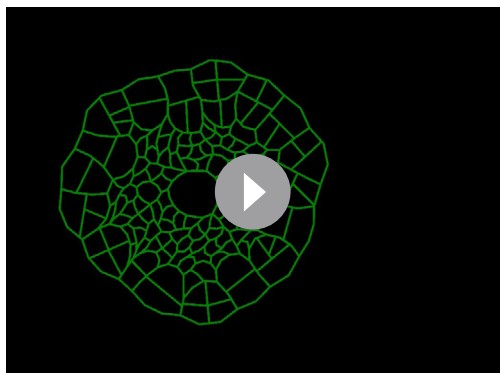

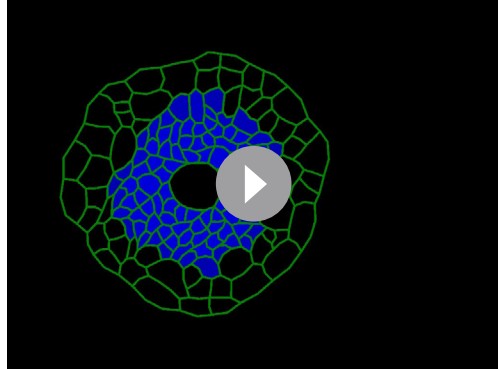

**Video 34.** Model 3C output, visualizing PXY<sub>active</sub>*.
https://elifesciences.org/articles/66627/figures#video34

**Video 35.** Model 3C output, visualizing PXY*.
https://elifesciences.org/articles/66627/figures#video35

cell* and drew lines between immediate daughter cells* (*Figure 5E and F*). We then calculated the goodness of fit ($R^2$) of a linear relationship between center of mass coordinates of all individual lines as a proxy for lineage* 'radiality' and, thus, for the ratio of periclinal versus anticlinal cell divisions*. After obtaining the $R^2$ value for each lineage*, we tested for median differences among r distributions under each stiffness regime (*Figure 5G*, *Figure 5—figure supplement 3*). These comparisons showed that the increase of the xylem* to non-xylem cell wall* stiffness ratio produced a shift from more 'curved' lineages ($R^2 < 0.25$) toward more radial lineages* ($R^2 > 0.75$) (*Figure 5G*, *Figure 5—figure supplement 3A*). We attributed this effect to an increased radial elongation of cambium cells* with increasing xylem stiffness* (*Video 42*, *Video 43*, *Video 44*, *Video 45*, *Video 46*, *Video 47*) and the preferred cell division* along the shortest axis in VirtualLeaf (*Merks et al., 2011*). Although the effect of xylem cell* stiffness on lineage radiality was not on all lineages, as a fraction of them remained less radially oriented even for high xylem stiffness (*Figure 5G*), implementing stiffness as a cell property therefore produced coherent results in terms of the appearance of radial cell files* as an emergent property of xylem cell* wall stiffness. In contrast, the analysis of different epidermis cell wall stiffness did not show a clear change in the distribution of lineages in the range of analyzed stiffness regimes (*Figure 5H*, *Figure 5—figure supplement 3B*, *Video 48*, *Video 49*, *Video 50*, *Video 51*, *Video 52*) as increasing stiffness limited tissue growth and therefore the formation of cell lineages. These results remained consistent for both xylem* and epidermis* stiffness regimes when varying other parameters determining cell wall dynamics, that is, the target length of cell wall elements and the yielding threshold for the introduction of new cell wall segments (*Figure 5—figure supplement 4*).

## Discussion

Growth and development of multicellular organisms are complex nonlinear processes whose dynamics and network properties are not possible to predict only based on information on their individual building blocks and their one-to-one interactions. The rather simple cellular outline along the radial axes of plant organs, growth in only two dimensions, and the recent identification of central functional properties (*Smetana et al., 2019*; *Miyashima et al., 2019*; *Shi et al., 2019*) make radial plant growth an attractive target for a systematic approach to reveal its intriguing dynamics. Here, we developed a computational model representing a minimal framework required for radial plant growth using the VirtualLeaf framework (*Merks et al., 2011*). In particular, we combined an agent-based model of the tissue layout with an ODE model of the inter-cellular PXY/CLE41 signaling module. By integrating these two modeling and biological scales, we were able to recapitulate not only the complex behaviors that arise as consequence of the cellular interactions (*Macal and North, 2005*) but also the interplay between cellular layout and intercellular signaling dynamics. Therefore, our model allows analyzing fundamental features of plant organ growth and integrates the PXY/CLE41 module as one central element for cambium patterning and maintenance.

Using positional information mediated by morphogenetic gradients of diffusible chemicals to pattern growing structures is a classical concept in developmental biology that has stirred a long

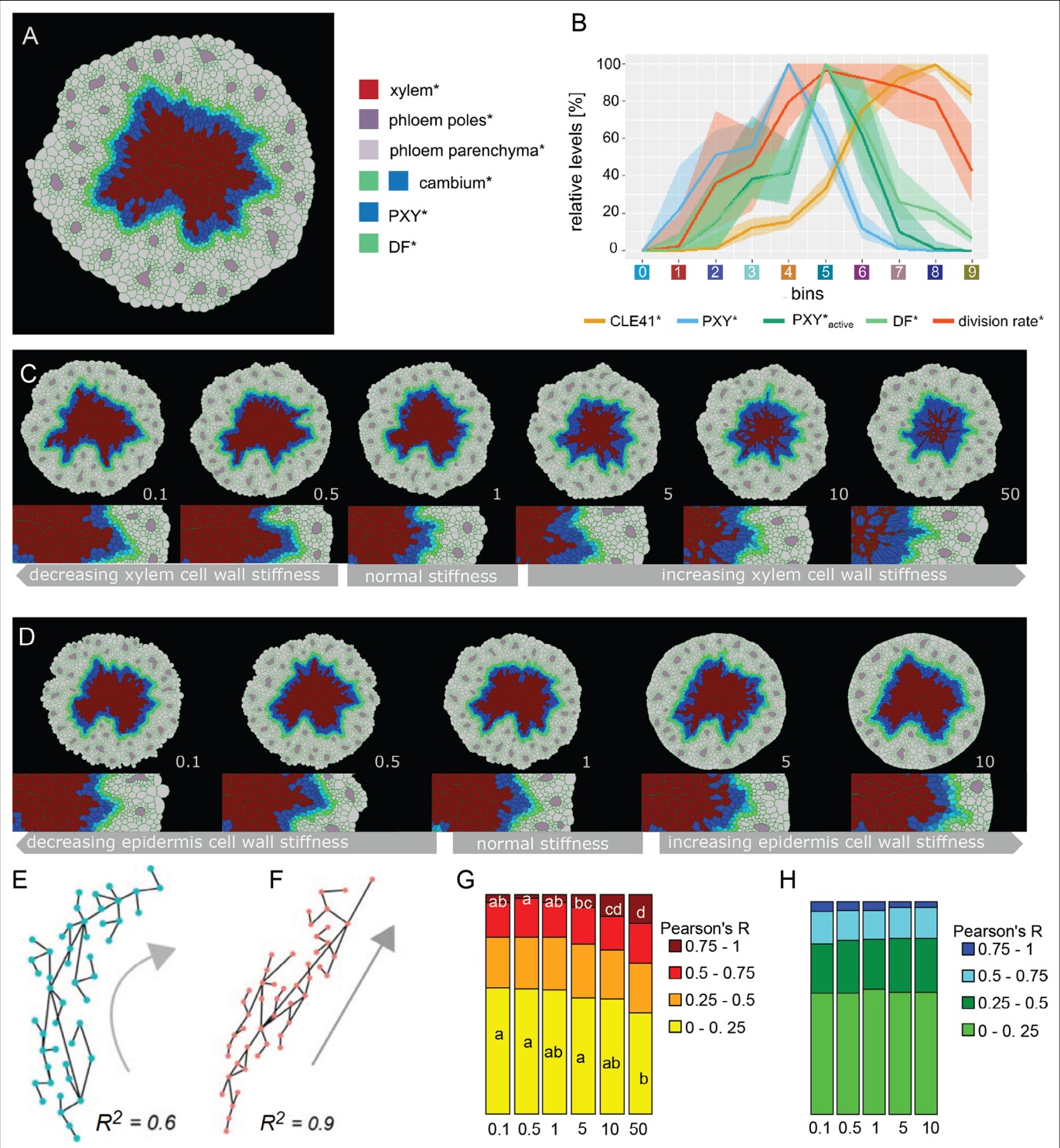

**Figure 5.** Effect of xylem cell wall stiffness* on the radiality of cambium-derived cell lineages*. (**A**) Final output of Model 4 and parameter set 1. (**B**) Visualization of the relative levels of chemicals* and division rates* in different bins. Bin colors along the x-axis correspond to the different bins similarly as in *Figure 4C*. The shading represents the range between minimal and maximal values during simulations (n = 10). (**C, D**) Simulation outputs at increasing values of xylem stiffness* (C)and epidermis stiffness* (D)with the ratio of stiffness* vs. experimentally determined xylem stiffness indicated at the right bottom corner of each example. All the simulations had the same starting conditions and ran for the same amount of simulated time. At the bottom, there is a magnification of the right region shown in the pictures above, respectively. (**E, F**) Examples of the relationship between $R^2$ and the geometry of proliferation trajectories (gray arrows) for two different $R^2$ values; dots are cell* centroids, lines represent division* events. (**G, H**) Fraction of median relative amount of lineages whose $R^2$ falls within a specific range for 30 simulations in each condition (n ≥ 70 lineages per simulation) at different

*Figure 5 continued on next page*

*Figure 5 continued*

xylem stiffness* (G)and epidermis stiffness* (H)regimes. In case of significant difference among medians, assessed with Kruskal–Wallis (KW significance is p<2.6 E-3 for (0, 0.25) interval and p<9.17e-7 for the (0.75,1) interval), the pairwise difference between medians was tested post hoc applying the Dunn test. The post hoc results are reported in each box as letters; medians sharing the same letter or do not display a letter at all do not differ significantly.

The online version of this article includes the following figure supplement(s) for figure 5:

**Figure supplement 1.** Distribution of cell* properties under different xylem 'stiffness' regimes.

**Figure supplement 2.** Distribution of cell* properties under different tissue boundary (=epidermis*) 'stiffness' regimes.

**Figure supplement 3.** Fraction of median relative amount of cell lineages for parameter sets 2–5 for n = 30 simulations and n ≥ 60 lineages.

**Figure supplement 4.** Fraction of median relative amount of cell lineages at different parameters governing cell wall* dynamics.

history of fundamental debates (*Green and Sharpe, 2015*). Initially, we used the PXY/CLE41 module to generate such a gradient instructing cambium cells* to differentiate into xylem cells*, to proliferate, or to differentiate into phloem cells*. Repression of cell division in the distal cambium was achieved by implementing an inhibitory feedback loop of PXY-signaling* on PXY* production. Altogether, this setup was already sufficient to maintain stable radial tissue organization during radial growth and established a maximum of cell division rates in the cambium center as observed by experimental means (*Shi et al., 2019*). Thus, we conclude that cambium organization and radial patterning of plant growth axes can be maintained by a distinct pattern of radially acting morphogens. Such a role was initially proposed for auxin whose differential distribution, however, seems to be rather a result of tissue patterning than being instructive for radial tissue organization (*Bhalerao and Fischer, 2014*).

In contrast to expected roles of the *PXY* pathway in xylem formation based on experiments during primary vascular development (*Hirakawa et al., 2008*; *Hirakawa et al., 2010*; *Kondo et al., 2014*), we observed that the overall amount of proximal tissue production during radial plant growth did not depend on the *PXY* function. Automated image analysis including object classification revealed that neither the number of cells produced toward the organ center nor the number of vessel elements did change in a *pxy* mutant background but rather the ratio between parenchyma and fiber cells. Therefore, in contrast to a negative effect of PXY/CLE41 signaling on vessel formation in vascular bundles in leaves (*Hirakawa et al., 2008*; *Kondo et al., 2014*), vessel formation during radial plant growth is *PXY/CLE41*-independent. Instead, fiber formation is positively associated with the *PXY/CLE41* module. These observations indicated that xylem formation is unlikely to be instructed by PXY/CLE41 signaling alone and that additional signals are required.

Moreover, the application of markers visualizing cambium organization showed that *PXY*-deficiency leads to cambium disorganization in some regions of the hypocotyl, whereas in other areas, cambium anatomy is maintained. Since such areas are regularly spaced, this pattern may arise due to factors acting in parallel to PXY/CLE41 and which also carry spatial information. Although ethylene signaling was reported to act in parallel to PXY/CLE41 signaling, spatial specificity does not seem to be a characteristic property of ethylene signaling (*Etchells et al., 2012*). In contrast, PEAR transcription

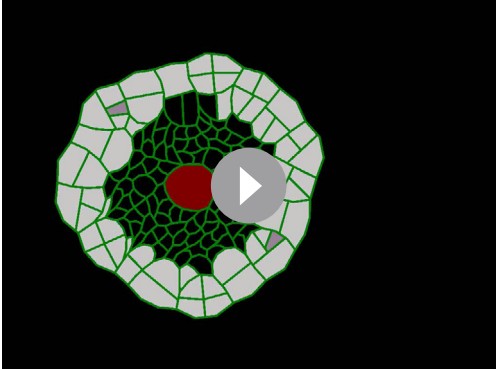

**Video 36.** Model 4 output, parameter Set 1, visualizing xylem (red), phloem parenchyma (light purple), and phloem poles (dark purple), and accumulation of PXY* (blue) and the division chemical (DF)* (green).

https://elifesciences.org/articles/66627/figures#video36

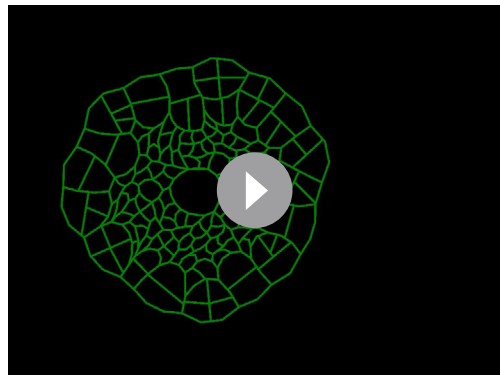

**Video 37.** Model 4 output, parameter set 1, visualizing CLE41* (yellow) accumulation.

https://elifesciences.org/articles/66627/figures#video37

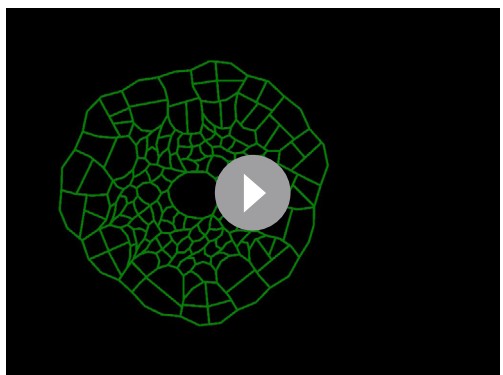

**Video 38.** Model 4 output, parameter set 1, cell divisions (red).

https://elifesciences.org/articles/66627/figures#video38

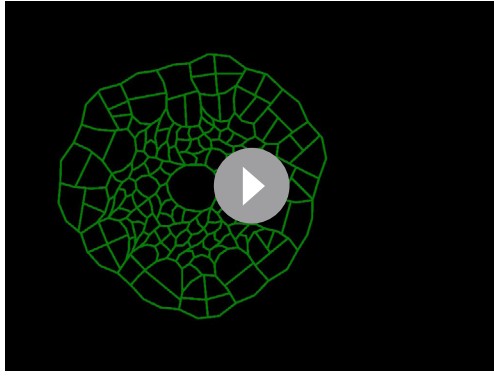

**Video 39.** Model 4 output, parameter set 1, visualizing accumulation of PXY* (blue) and the division chemical (DF)* (green).

https://elifesciences.org/articles/66627/figures#video39

factors are phloem-derived and stimulate the proliferation of cambium stem cells presumably in a PXY/CLE41-independent manner (*Miyashima et al., 2019*) and, thus, may act similarly to the PF* factor we introduced in our model. The ERECTA/EPIDERMAL PATTERNING FACTOR-LIKE (ER/EPFL) receptor-ligand pathway acting in concert with the PXY/CLE41 module (*Wang et al., 2019*; *Etchells et al., 2013*) represents another candidate for playing such a role. In addition, CLE45 was recently proposed to be expressed in developing sieve elements, the conducting units of the phloem, and repress the establishment of sieve element identity in their immediate environment mediated by the RPK2 receptor protein (*Gujas et al., 2020*). The PF* factor in our model combines features of these phloem-derived molecules.

In addition to the phloem sending out instructive signals, early xylem cells have been identified to act as an organizing center of cambium patterning (*Smetana et al., 2019*). Although this finding seems to be at odds with our claim that phloem-derived signals are sufficient for cambium organization, it is important to consider that we ignored, for example, upstream regulation of postulated factors like PXY* or CLE41*, which obviously depends on positional information which could be mediated in plants by auxin or cytokinin signaling (*Bishopp et al., 2011*). For simplicity, we also ignored organizing effects of signaling longitudinally to cross-sections as it can, for example, be expected for polar auxin transport (*Bennett et al., 2016*; *Ibañes et al., 2009*; *Fàbregas et al., 2015*) in the context of cambium activity or xylem formation. Although being considerably more complex, the establishment of 3D models will be crucial and essential for addressing this aspect.

In this context, it is interesting to note that we deliberately excluded the transition from the initially bisymmetric tissue conformation to a concentric tissue organization as it occurs in hypocotyls and roots (*Smetana et al., 2019*; *Sankar et al., 2014*)

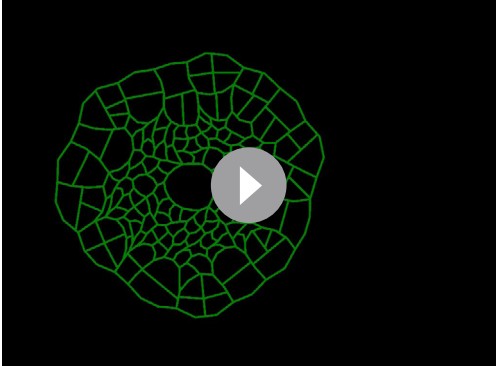

**Video 40.** Model 4 output, parameter set 1, visualizing $PXY_{active}$*.

https://elifesciences.org/articles/66627/figures#video40

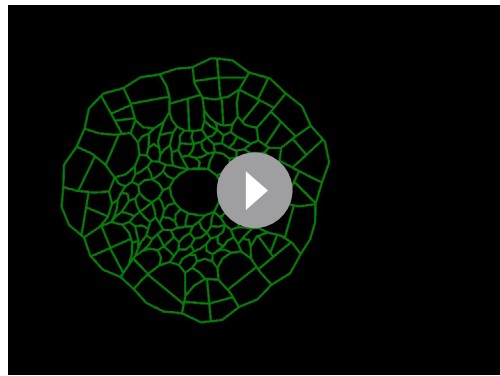

**Video 41.** Model 4 output, parameter set 1, visualizing PXY*.

https://elifesciences.org/articles/66627/figures#video41

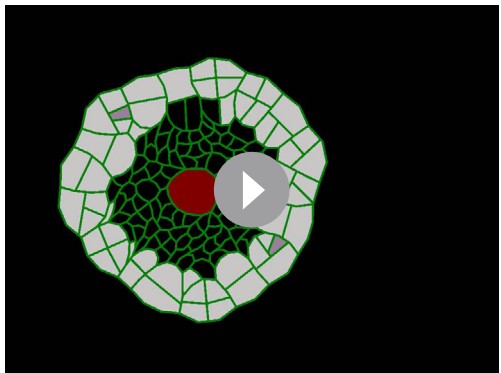

**Video 42.** Model 4 output, visualizing accumulation of PXY* (blue) and the division chemical (DF)* (green) implementing a 0.1-fold change in xylem* cell wall stiffness.
https://elifesciences.org/articles/66627/figures#video42

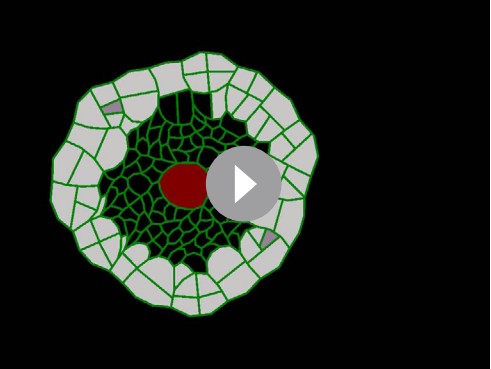

**Video 43.** Model 4 output, visualizing accumulation of PXY* (blue) and the division chemical (DF)* (green) implementing a 0.5-fold change in xylem* cell wall stiffness.
https://elifesciences.org/articles/66627/figures#video43

from our model. Our rationale was that the rather complex change in tissue anatomy from a primary to a secondary conformation in the hypocotyl required more assumptions in our model and would have spoiled the advantages of a relatively simple anatomy for generating a cell-based computational model. Moreover, the differences in primary anatomy of shoots and roots before the onset of radial plant growth (*Smetana et al., 2019*; *Sehr et al., 2010*) would have required different cellular outlines for both cases and, thus, would have hampered the generality of our approach.

Interestingly, the front of cambium domains is very stable, that is, almost perfectly circular, in planta but this is not the case for our computational simulations. We believe that instability in the computational models is due to local noise in the cellular pattern leading to differential diffusion of chemicals* with respect to their radial position and to a progressive deviation of domains from a perfect circle. Such a deviation seems to be corrected by an unknown mechanism in planta but such a corrective mechanism is due to the absence of a good indication in planta not implemented in our models. Analyses of wt and *pxy* lines at different stages (*Figure 3—figure supplement 1*, *Figure 3—figure supplement 2*, *Figure 3—figure supplement 3*, *Figure 3—figure supplement 4*) revealed 'gaps' in the cambium domain already at early stages of *pxy* development arguing against the possibility that the *pxy* anatomy is caused by increased front instability. Although a corrective mechanism ensuring front stability in planta is difficult to predict, our model now allows to test respective ideas like directional movement of chemicals or stabilizing communication between cells during cambium activity.

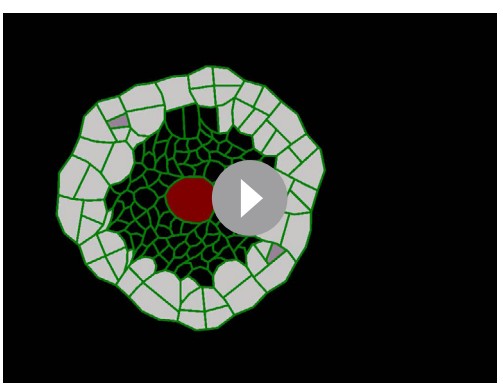

**Video 44.** Model 4 output, visualizing accumulation of PXY* (blue) and the division chemical (DF)* (green) at experimentally determined xylem cell wall stiffness.
https://elifesciences.org/articles/66627/figures#video44

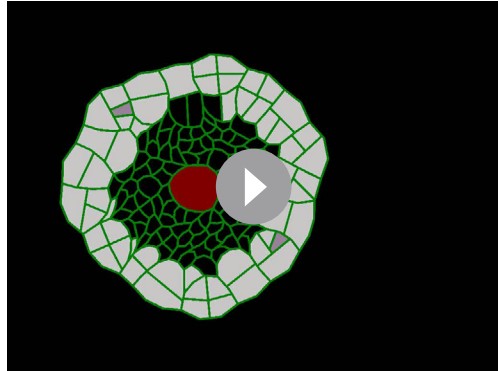

**Video 45.** Model 4 output, visualizing accumulation of PXY* (blue) and the division chemical (DF)* (green), implementing a 5-fold increase in xylem* cell wall stiffness.
https://elifesciences.org/articles/66627/figures#video45

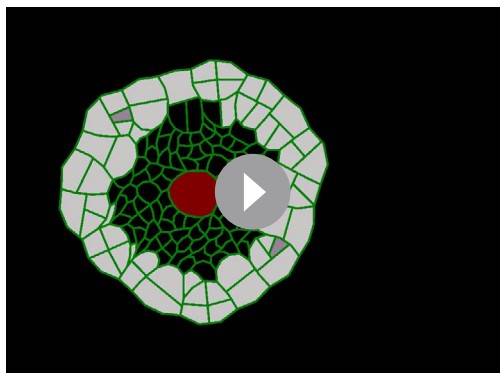

**Video 46.** Model 4 output, visualizing accumulation of PXY* (blue) and the division chemical (DF)* (green), a tenfold increase in xylem* cell wall stiffness.
https://elifesciences.org/articles/66627/figures#video46

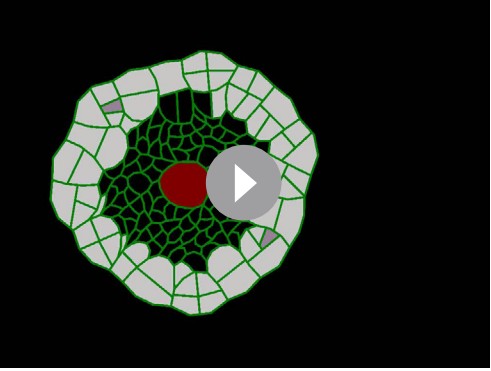

**Video 47.** Model 4 output, visualizing accumulation of PXY* (blue) and the division chemical (DF)* (green), a 50-fold increase in xylem* cell wall stiffness.
https://elifesciences.org/articles/66627/figures#video47

In this context, it is interesting that increasing epidermis* 'stiffness' increased circularity of the growing organ*, which may be administered by the periderm (*Serra et al., 2022*), the protective cell layers that we did not consider in our model.

Current research on plant mechanical biology indicates how cell mechanical properties influence cell and tissue morphogenesis. Microtubules, turgor pressure, and cell wall composition are central factors in this regard (*Sapala et al., 2018*; *Altartouri et al., 2019*). Due to the geometric constraints in a radially growing plant axis, it becomes challenging to uncouple these factors experimentally and establish the impact of one factor on organ patterning during radial plant growth. By expanding VirtualLeaf to allow for the integration of cell-type-specific wall stiffness, we fundamentally increased the spectrum of potential modeling approaches. In particular, since cell wall stiffness is accessible by the cellular model throughout simulations, it is now possible to simulate and analyze, for example, the dynamics of auxin or brassinosteroid-mediated cell wall loosening (*Majda and Robert, 2018*; *Caesar et al., 2011*). In our cambium model, by modulating exclusively cellular 'stiffness,' we were able to computationally simplify the 'physical' properties and, thereby, develop a hypothesis how inter-tissue forces influence stem cell behavior not only cell autonomously, but also in a non-cell-autonomous manner.

Taken together, we envision that the model presented in this study recapitulates the qualitative and quantitative variation in radial plant growth on multiple levels, found in different mutants or when comparing different dicotyledonous species (*Spicer and Groover, 2010*). Remarkable features like the establishment of concentric cambium rings often found in the order of *Caryophyllales* (*Carlquist,*

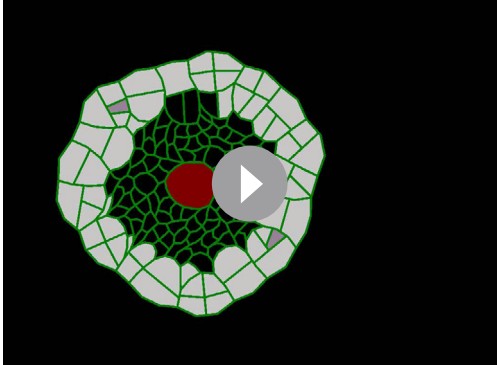

**Video 48.** Model 4 output, visualizing accumulation of PXY* (blue) and the division chemical (DF)* (green) implementing a 0.1-fold change in epidermis* cell wall stiffness.
https://elifesciences.org/articles/66627/figures#video48

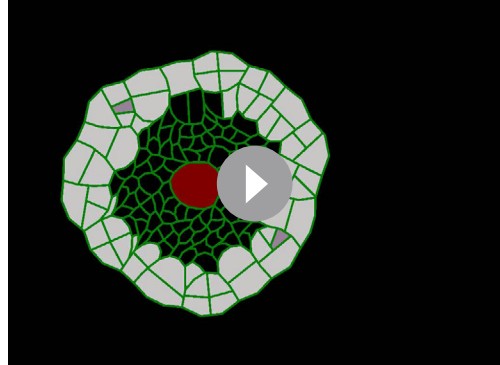

**Video 49.** Model 4 output, visualizing accumulation of PXY* (blue) and the division chemical (DF)* (green) implementing a 0.5-fold change in epidermis* cell wall stiffness.
https://elifesciences.org/articles/66627/figures#video49

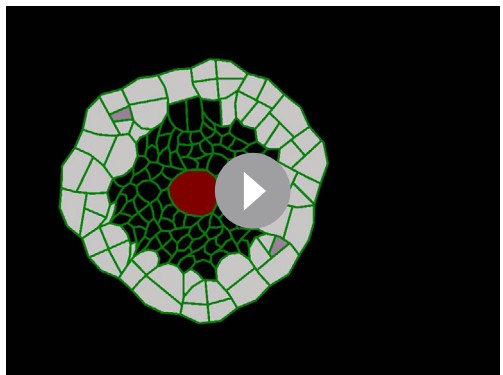

**Video 50.** Model 4 output, visualizing accumulation of PXY* (blue) and the division chemical (DF)* (green) at experimentally determined epidermis* cell wall stiffness.

https://elifesciences.org/articles/66627/figures#video50

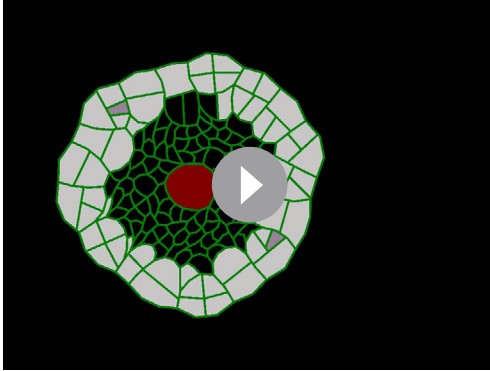

**Video 51.** Model 4 output, visualizing accumulation of PXY* (blue) and the division chemical (DF)* (green), implementing a fivefold increase in epidermis* cell wall stiffness.

https://elifesciences.org/articles/66627/figures#video51

*2007*) or 'phloem wedges' found in the *Bignonieae* genus (*Pace et al., 2009*) may be recapitulated by adjusting the model's parameters values or by introducing additional factors. In the future, the model may help to predict targets of environmental stimuli inducing changes of cambium activity like seasonal changes (*Bhalerao and Fischer, 2017*) or mechanical perturbation (*Gerttula et al., 2015*), allowing the generation of testable hypotheses. Thus, our dynamic model will be a useful tool for investigating a process not possible to observe in real time and partly develops over exceptionally long periods.

## Materials and methods
### Plant material and growth conditions

*Arabidopsis thaliana* (L.) Heynh. plants of Columbia-0 accession were used for all experiments and grown as described previously (*Suer et al., 2011*). *pxy-4* (SALK_009542, N800038) mutants were ordered from the Nottingham Arabidopsis Stock Centre (NASC). Plant lines carrying *IRX3pro:CLE41* and *35Spro:CLE41* transgenes (*Etchells and Turner, 2010*) were kindly provided by Peter Etchells (Durham University, UK). *PXYpro:ECFP-ER* (*pPS19*) and *SMXL5pro:EYFP-ER* (*pJA24*) reporter lines expressing fluorescent proteins targeted to the endoplasmatic reticulum (ER) were described previously (*Agusti et al., 2011a*; *Wallner et al., 2017*). After sterilization, seeds were stratified for 2–3 d in darkness at 4°C. Plants were then grown at 21°C and 60% humidity. To check *PXYpro:CFP/SMXL-5pro:YFP* activities, 27- or 39-d-old seedlings were used. 27-d-old seedlings were grown on plates in short-day conditions (10 hr light and 14 h darkness). 39-d-old seedlings were grown on soil in short-day conditions for 21 d and then moved to long-day conditions (16 hr light and 8 hr darkness) for 18 d.

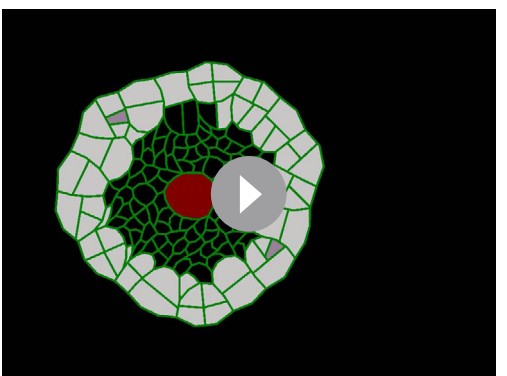

**Video 52.** Model 4 output, visualizing accumulation of PXY* (blue) and the division chemical (DF)* (green), a tenfold increase in epidermis* cell wall stiffness.

https://elifesciences.org/articles/66627/figures#video52

### Confocal microscopy

Hypocotyls were isolated and cleaned from surrounding leaf material using razor blades (Classic Wilkinson, Germany). The cleaned hypocotyls were mounted in 7% low melting point agar (Sigma-Aldrich, St. Louis, MO) in water and sections were generated using a vibratome (Leica VT1000 S). For monitoring hypocotyl development, the developmental gradient in hypocotyls of 27-d-old plate-grown plants (stages 1–3 shown in *Figure 3—figure supplement 1A–C, E–G, I–K*

and *Figure 3—figure supplement 2A–C, E–G, I–K*) was employed: the lower region close to the hypocotyl-root boundary was taken as the youngest stage (stage1), the middle region as stage 2, and the upper region close to the cotyledons as stage 3. As stage 4, sections from the middle region of 39-d-old plants grown on soil were taken, which are shown in all other images displaying confocal analyses. For *Figure 4—figure supplement 2A and B*, 75-μm-thick sections from soil-grown 32-d-old plants were stained for 60 min with 0.1% w/v Direct Yellow 96 (Sigma-Aldrich, S472409-1G) diluted in ClearSee (*Ursache et al., 2018*) (10% w/v xylitol, 15% w/v sodium deoxycholate, 25% w/v urea), washed three times with ClearSee and mounted in ClearSee on microscope slides. For other experiments, 60-μm-thick sections were stained for 5 min with 0.1% w/v Direct Red 23 (Sigma-Aldrich, 212490-50G) diluted in water, washed three times with water, and mounted in water on microscope slides. For analyzing the fluorescent markers, a Leica SP8 or Stellaris 8 (Leica, Germany) confocal microscope was used. Different fluorescence protein signals were collected in different tracks. YFP was excited at 514 nm, and emission was collected at 522–542 nm. CFP was excited at 458 nm, and the signal emission was collected at 469–490 nm. The Direct Red 23-derived signal was excited at 495 nm, and emission was detected at 558–649 nm. The Direct Yellow 96-derived signal was excited at 488 nm, and emission was detected at 500–540 nm. For qualitative comparisons, 5–10 samples for each sample type were included and repeated at least twice. Please be aware that depending on variations in staining intensity, sometimes cell walls of vessel elements appear white in the provided images due to the overlap of signal from Direct Red 23 staining and autofluorescence captured during *PXYpro:CFP* detection (e.g., see *Figure 3—figure supplement 2D*).

## Ilastik cell-type counting

For cell-type classification and quantification, sections were produced from 42-d-old plants as previously described (*Suer et al., 2011*). The xylem area was cropped manually from histological images of wild-type and *pxy* mutant. The Ilastik toolkit (*Sommer et al., 2011*) was used for image segmentation and cell-type classification (https://www.ilastik.org). With a training set, the pixel classification workflow was trained to distinguish cell walls from the background. After segmentation, the object classifier was then trained to split the resulting objects into four groups – xylem vessels, xylem fibers, xylem parenchyma, and unclassified objects. The resulting classifier was then applied to all cropped images. For each image, cell-type data were extracted using Python. 11–12 plants each for wild-type and *pxy* mutants were compared in two independent experiments.

## VirtualLeaf simulations

Simulations were performed as recommended previously (*Merks et al., 2011*). To be able to see established models in action, the VirtualLeaf software was installed according to the following instructions described in Appendix 1 and as described previously (*Merks and Guravage, 2013*). All simulations within Models 1–4, respectively, were conducted for the same VirtualLeaf time duration and repeated at least 10 times to account for the stochastic nature of the tissue simulations (for details on simulations in VirtualLeaf, see section 'Description of the VirtualLeaf simulations' in Appendix 1). Dilution of the modeled variables due to growth has been omitted.

## Splitting the result of VirtualLeaf simulations into bins

After a VirtualLeaf simulation was completed, the resulting xml template was stored. To analyze the distribution of chemicals* in such a template along the radial axis, we produced a Python script named "Cambium_bins_calculation.ipnb." Within the script, it was possible to indicate the path to the xml file, and the script produced two.csv files – one with a table containing data about each cell and another with information about averages across the requested bin number. Cells were sorted into bins based on the cells' Euclidean distance from the center of the tissue, which was defined as the average of the x- and y-coordinates of all the cells in the tissue.

## Parameter estimation and exploration of the parameter space

To estimate the model parameters and, at the same time, investigate the parameter space, we performed a large set of simulations with randomized parameters to identify feasible parameter combinations. In particular, we employed a combination of Python and shell scripting to set up the parameter sets, run the simulations, and analyze the results. To generate the parameter sets, we

followed the tutorial using the Python library xml.etree.Elementree as described (*Antonovici et al., 2022*). The search intervals were defined based on the manually determined parameter values of Model 3A: the search interval was set between 1/3 and 3 times the original value. The individual parameter sets were then simulated for a duration of t_simulated = 2,200 steps on a computing cluster (Linux, 64-bit). The resulting xml leaf was then analyzed based on tissue size and proportions. Based on in planta observations (*Sankar et al., 2014*), we determined that the simulation should result in 24% cambium, 10% xylem, and 65% phloem cells. As all tissues are equally important, we used a weighted least-squares scoring function to compare the experimentally measured tissue ratios with the model simulations. We added a term for the total number of cells to favor parameter sets that resulted in tissue growth. Altogether, this resulted in the following scoring function:

$$x = \frac{1}{0.01}\left(0.1 - fraction_{xylem}\right)^2 + \frac{1}{0.05676}\left(0.24 - fraction_{cambium}\right)^2 + \frac{1}{0.4225}\left(0.65 - fraction_{phloem}\right)^2 + \left(1 - totalcells/3000\right)^2$$

As we were interested in obtaining simulations with an active cambium, we discarded simulations that resulted in hypocotyls* with less than 300 cells* in total and with cambium cells less than 30. We further eliminated any parameter sets with pronounced edge instability.

## Exploration of stiffness

To explore the effects of stiffer (i.e., less flexible) xylem cell walls and epidermis cell walls as represented by the perimeter stiffness, we slightly modified the VirtualLeaf code so that it was possible for $\lambda_L$ (the 'cost' of deviation of the wall element's length from the target length) to assume cell-type-specific values. More specifically, we defined a new parameter named *cellwallstiffness*, and set $\lambda_L$ = *cellwallstiffness* according to the experimentally determined cell wall thickness as a proxy for cell wall stiffness. We then ran the model with different ratios of *cellwallstiffness* compared to the normal parameter value, while maintaining the same tissue configuration used for the other simulations done within this study. The values chosen for the parameter were 0.1-, 0.5-, 1-, 5-, and 10-fold change for both tissues of interest and 50-fold change for xylem*. We replicated each run 30 times. We further repeated the analysis of different stiffness regimes while varying other cell wall dynamic parameters by ±50%, that is, the target element for cell wall elements and the yielding threshold for the introduction of new cell wall elements (for n = 10 simulations per parameter combination).

To study the proliferation trajectory of cells, we performed for every lineage a linear regression of the centers of mass for the cells belonging to that lineage and used the coefficient of determination ($R^2$) as proxy for proliferation trajectory of the lineage. We next tested for median differences among the $R^2$ distribution under each stiffness regime using the Kruskal–Wallis (KS) test, and performed the Dunn test to determine differences among groups in case of significant KS. Before performing the KS, we subsampled the data to maintain the same number of samples across stiffness values and bootstrapped the samples to obtain robust median estimators and confidence intervals.

Statistical analyses and visualizations of 'stiffness' were performed using the R language for statistical computing and graphics (https://www.r-project.org/) using the tidyverse family of packages (*Wickham et al., 2019*), together with the broom (*Robinson et al., 2023*), FSA (*Ogle et al., 2023*), and boot packages (*Davison and Hinkley, 1997*; *Canty and Ripley, 2020*).

## Acknowledgements

We thank Peter Etchells (Durham University, UK) for providing seed material, Karin Grünwald and Martina Laaber-Schwarz (both GMI, Vienna, Austria) for technical assistance and Dongbo Shi, Eva-Sophie Wallner, and Vadir López-Salmerón for comments on the experimental strategy and the manuscript. We also thank Claudiu Antonovici (University of Leiden, The Netherlands) for help in setting up the VirtualLeaf platform. This work was supported by the Deutsche Forschungsgemeinschaft (DFG) through the Research Unit FOR2581 'Plant Morphodynamics (grant GR2104/6), grant GR2104/4-1 and a Heisenberg Professorship (GR2104/5-2) to TG. RMHM was supported by Prof. Dr. Jan van der Hoevenstichting voor Theoretische Biologie. The work by BHM was initiated at Centrum Wiskunde & Informatica (CWI), Amsterdam, The Netherlands. RMHM and BHM thank CWI for providing a CWI Internship to BHM and for hosting IL. RMHM and RG thank the Mathematical Institute of Leiden

University for hosting RG. RG was supported by the CRC 1101 'Molecular Encoding of Specificity in Plant Processes' (DFG) and the Joachim Herz Stiftung.

## Additional information

### Funding

| Funder | Grant reference number | Author |
|---|---|---|
| Deutsche Forschungsgemeinschaft | GR2104/4-1 | Thomas Greb |
| Deutsche Forschungsgemeinschaft | GR2104/5-2 | Thomas Greb |
| Deutsche Forschungsgemeinschaft | GR2104/6 | Thomas Greb |
| Joachim Herz Stiftung | | Ruth Großeholz |
| Deutsche Forschungsgemeinschaft | CRC 1101 | Ruth Großeholz |

The funders had no role in study design, data collection and interpretation, or the decision to submit the work for publication.

### Author contributions

Ivan Lebovka, Conceptualization, Resources, Software, Formal analysis, Investigation, Visualization, Methodology, Writing - original draft; Bruno Hay Mele, Conceptualization, Software, Writing – review and editing; Xiaomin Liu, Theresa Schlamp, Investigation; Alexandra Zakieva, Formal analysis, Investigation, Methodology, Writing – review and editing; Nial Rau Gursanscky, Supervision; Roeland MH Merks, Conceptualization, Resources, Supervision, Funding acquisition, Project administration, Writing – review and editing; Ruth Großeholz, Conceptualization, Data curation, Software, Formal analysis, Funding acquisition, Investigation, Methodology, Writing – review and editing; Thomas Greb, Conceptualization, Supervision, Funding acquisition, Visualization, Project administration, Writing – review and editing

### Author ORCIDs

Ivan Lebovka http://orcid.org/0009-0002-7721-0738
Bruno Hay Mele http://orcid.org/0000-0001-5579-183X
Xiaomin Liu http://orcid.org/0000-0002-8592-5752
Alexandra Zakieva http://orcid.org/0000-0002-6089-3766
Theresa Schlamp http://orcid.org/0000-0002-2986-5134
Roeland MH Merks http://orcid.org/0000-0002-6152-687X
Ruth Großeholz http://orcid.org/0000-0003-4604-4538
Thomas Greb http://orcid.org/0000-0002-6176-646X

### Decision letter and Author response

Decision letter https://doi.org/10.7554/eLife.66627.sa1
Author response https://doi.org/10.7554/eLife.66627.sa2

## Additional files

### Supplementary files

• Supplementary file 1. Table listing cell* behavior rules for Models 1–4.

• Supplementary file 2. Table listing parameter values and chemical thresholds after parameter estimation.

• Transparent reporting form

### Data availability

Code files for presented models are deposited at https://github.com/thomasgreb/Lebovka-et-al_cambium-models (copy archived at *Lebovka et al., 2023*).

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

## Appendix 1

### Description of the VirtualLeaf simulations

VirtualLeaf allows for models to combine tissue dynamics, cell behavior dynamics, and biochemical networks that span between cells. The different modeling scales are simulated iteratively: During each simulation step, the tissue dynamics are simulated first using Monte Carlo simulations until a stable energy of the Hamiltonian is reached. Only then are the biological rules applied, with cell division occurring last in order to prevent new cells from interfering with the simulations.

For a detailed description of the simulation process, see *Merks et al., 2011*; *Merks and Guravage, 2013*, and *Antonovici et al., 2022*. Here, we include a brief overview of tissue simulations in VirtualLeaf and outline the changes we made for Model 4 as well as the biological rules of the different cambium model versions. The base VirtualLeaf source code is available for download from https://github.com/rmerks/VirtualLeaf2021, (*Merks et al., 2021*). The custom version of VirtualLeaf that we built for this analysis as well as the models described in this article are available at https://github.com/thomasgreb/Lebovka-et-al_cambium-models, (copy archived at *Lebovka et al., 2023*).

### Tissue simulations

The tissue dynamics are simulated using Monte Carlo simulation dynamics. Briefly, VirtualLeaf attempts to move all nodes of the model in a random order. A Hamiltonian operator is used to assess the energy of the system at both the old and the new position of the node. The movement of nodes is accepted if it minimizes the energy of the system. This operator considers both the cells' compression and the resistance of the cell wall elements to being stretched or compressed (*Merks et al., 2011*):

$$H = \lambda_A \sum_i \left( a\left(i\right) - A_T\left(i\right) \right)^2 - \lambda_M \sum_j \left( l\left(j\right) - L_T\left(j\right) \right)^2$$

with $\lambda_A$ as the cell's resistance to compression or expansion, $\lambda_M$ the spring constant for the cell wall elements, $A_T$ and $L_T$ are the cell's target area and the cell wall's target length, respectively, with $a\left(i\right)$ representing the current cell area and $l\left(j\right)$ the current wall length.

Cellular growth is implemented in VirtualLeaf as an increase in the cells' target areas. Until the maximal cell size is reached, a cell's target area $A_T\left(i\right)$ is increased by a fixed amount in each simulation step. This results in increasing the contribution of the area compression to the Hamiltonian operator

For Model 4, the calculation of the Hamiltonian was refined to include a more detailed definition of the second term for the calculation of the cell wall component of the system's energetic state:

$$\lambda_M \sum \frac{\left(\lambda_{L1} + \lambda_{L2}\right)}{2} \left( l\left(j\right) - L_T\left(j\right) \right)^2$$

Here, $\lambda_{L1,2}$ are cell-specific spring constants for the cells that share each specific wall element $j$. Specifically, $\lambda_{L1}$ and $\lambda_{L2}$ are relative contributions to the stiffness of the joint cell wall, where each contribution represents the half of the cell wall secreted by that particular cell. To make the cell wall module compatible with earlier VirtualLeaf models, the default value for $\lambda_L$ is set to '1' such that the expanded calculations result in a multiplication by '1' and do not affect the calculations of the Hamiltonian. For Models 1 - 3C the default values were used such that the standard Hamiltonian Operator applied. The changes to the code in our custom version of VirtualLeaf are marked by a comment 'Lebovka et al.' at the respective lines of code.

Take the cellular layout of *Figure 1* as exemplary situation, where node 5 is being moved. During the calculations of the cell wall elements, there are three walls to consider: between nodes 5 and 6, between 5 and 7, and between 5 and 4. As indicated by the arrows, each cell wall will be considered twice during the calculations for the move of node 5: the cell wall between node 4 and node 5 will be called once for cell 1 and once for cell 3, taking into account the specific cell wall thickness specific for each cell.

Altogether, this allows a cell-typespecific representation of the stiffness of the cell wall elements and therefore a more realistic representation of tissue structure such as an increased cell wall thickness and stability of xylem cells.

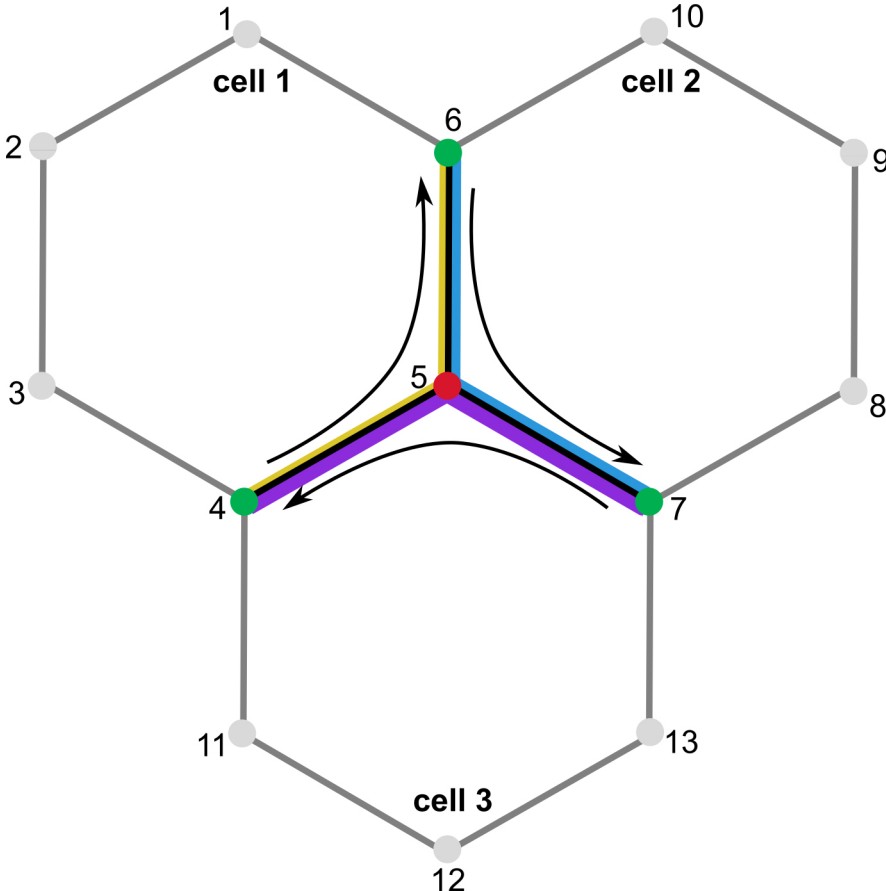

**Appendix 1—figure 1.** Cell wall calculations during node movement. Node 5 is moved to a new position. During calculations, the change in wall elements between nodes 5 and 6, 5 and 7, as well as 5 and 4 is considered. The cell-specific stiffness of the wall elements is indicated by the thickness of the colored lines – yellow for cell 1, blue for cell 2, and purple for cell 3.

As all the nodes are moved in a random order, this may cause some variation on the tissue layout between simulations. As a consequence, the application of cell behavior rules can vary as well between simulation runs, for example, as cells divide over the longest axis and not in a predetermined direction. To account for these variations between simulations, we simulated each model, each parameter set, and each thickness regime at least ten times.

## Cambium models

Models in VirtualLeaf comprise four different files: (1) the project file, (2) the model header file, (3) the C++file containing the model algorithm, and (4) the tissue layout. We provide these four files for the cambium models in the GitHub repository linked above. The model will further need to be included in the Model.pro file as a subdirectory by including the line 'Model_folder \' as one of the entries below 'SUBDIR = \'.

### Model.pro
This is a C++project file containing the configuration settings and pathways for the necessary directories.

### Model.h
This is a C++header file containing a line with the following structure: 'virtual QString DefaultLeafML(void) {return QString("hypo7.xml");}.' The line indicates where VitualLeaf should search for an xml file that describes the structure of the tissue template (called 'leaf') used for the model to run upon. In this particular example, the name of the xml template is 'hypo7.xml.' VirtualLeaf will go to the folder in which you installed the software and will look for this file in the

subfolder './data/leaves.' In our case, a Windows machine was used. Therefore, the full path looked like this: 'C:\VirtualLeaf2021-main\data\leaves' and this folder contained a file 'hypo7.xml.' Please note that paths will be different depending on the operating system being used.

## Model.cpp

A C++file containing the model algorithm to reproduce the output described in this study. Each model contains specific rules for cell behavior and biochemical equations specific to the cell types defined in the leaf.xml file. The cell behavior rules are listed in the sections *OnDivide* and *CellHousekeeping* while the biochemical model is listed in the section *CellDynamics*. Cell-to-cell transport is considered in the section *CelltoCellTransport* with reactions at cell walls having their specific section *WallDynamics*, though the latter was not used in any of the Cambium models.

## Cell behavior models

All cells in the cambium models follow specific behavioral rules governing cell growth, cell division, and cell differentiation (*Supplementary file 2*). Generally, cells grow until a maximal size is reached unless other behavior rules are triggered. Cell division and differentiation require not only a minimal cell size but also additional conditions regarding chemical concentrations. Unless otherwise specified, all cell behavior rules are applied as long as the specific conditions are met.

## Biochemical model

### Models 1 and 2

In cambium* and xylem* cells, CLE41 dynamics are a combination of the diffusion of CLE41*, the binding to PXY* and the degradation of CLE41*:

$$\frac{d}{dt}\left[CLE41^*\right] = diffusion_{CLE41} - \left[PXY^*\right] \cdot \left[CLE41^*\right] - degradation_{CLE41} \cdot \left[CLE41^*\right]$$

In phloem* cells, there is an additional term in the equation describing the production of CLE41*:

$$\frac{d}{dt}\left[CLE41^*\right] = diffusion_{CLE41} + production_{CLE41} - \left[PXY^*\right] \cdot \left[CLE41^*\right]$$

$$-degradation_{CLE41} \cdot \left[CLE41^*\right]$$

PXY* is produced in cambium* cells and negatively regulated by bound PXY*:

$$\frac{d}{dt}\left[PXY^*\right] = \frac{production_{PXY}}{\left(1 + suppressrate \cdot \left[PXY_{active}\right]\right)} - \left[PXY^*\right] \cdot \left[CLE41^*\right]$$

$$-degradation_{PXY} \cdot \left[PXY^*\right]$$

In the other cell types* in turn, free PXY* is governed by CLE41* binding to PXY* as well as the degradation of the receptor:

$$\frac{d}{dt}\left[PXY^*\right] = -\left[PXY^*\right] \cdot \left[CLE41^*\right] - degradation_{PXY} \cdot \left[PXY^*\right]$$

The ODE describing the dynamics of bound PXY* is identical for all cell types*. Here, bound PXY* is produced by the association of CLE41* and PXY and later degraded:

$$\frac{d}{dt}\left[PXY^*_{active}\right] = \left[PXY^*\right] \cdot \left[CLE41^*\right] - degradation_{PXY^*_{active}} \cdot \left[PXY^*_{active}\right]$$

### Model 2B

In model 2B, CLE41* is also produced in xylem cells*, such that the ODE now reads as follows:

$$\frac{d}{dt}\left[CLE41^*\right] = diffusion_{CLE41} + production_{CLE41} - \left[PXY^*\right] \cdot \left[CLE41^*\right]$$

$$-degradation_{CLE41} \cdot \left[CLE41^*\right]$$

## Model 2C and D

In Models 2C and D, the production of PXY* in cambium cells is eliminated (C) or strongly reduced (D). As such, the ODE for PXY* in model 2D is now

$$\frac{d}{dt}\left[PXY^*\right] = \frac{0.1 \cdot production_{PXY}}{\left(1 + suppressrate \cdot \left[PXY_{active}\right]\right)} - \left[PXY^*\right] \cdot \left[CLE41^*\right]$$

$$-degradation_{PXY} \cdot \left[PXY^*\right]$$

For Model 2C, the production term is set to '0,' fully eliminating PXY* production in cambium cells*.

## Models 3 and 4

In Models 3 and 4, we expanded the biochemical network to include additional chemicals suppressing PXY expression (RP*), a dedicated division factor, as well as phloem-derived factors promoting the division factor and suppressing phloem pole formation (PF$_{div}$* and PF$_{pole}$*, respectively). While the ODEs for CLE41*, free PXY* and bound PXY* remain mostly unchanged, we refined the ODE for PXY* to make the production of PXY* independent of PXY$_{active}$*:

$$\frac{d}{dt}\left[PXY^*\right] = \frac{production_{PXY}}{\left(1 + suppressrate \cdot \left[RP^*\right]\right)} - \left[PXY^*\right] \cdot \left[CLE41^*\right]$$

$$-degradation_{PXY} \cdot \left[PXY^*\right]$$

We also set the production rates of CLE41* to be higher in phloem poles* than in phloem parenchyma*.

The factor suppressing PXY expression (RP*) diffuses and is degraded throughout the tissue but is only produced in phloem cells. We therefore get in the following equation for phloem cells:

$$\frac{d}{dt}\left[RP^*\right] = production_{RP} + diffusion_{RP} - degradation_{RP} \cdot \left[RP^*\right]$$

In all other cell types, this ODE is simplified to include only the diffusion and degradation of RP*.

For the second phloem-derived factor, PF*, two chemicals were defined in the biochemical model on account of the different functions in the model reminiscent of different signaling components in planta: promoting the production of the division chemical reminiscent of the PEAR transcription factors (PF$_{div}$*) and suppressing phloem pole formation reminiscent of the CLE45/RPK2 signaling module (PF$_{pole}$*). The respective ODEs for both PF$_{div}$* and PF$_{pole}$* in phloem poles* are therefore:

$$\frac{d}{dt}\left[PF_{div}^*\right] = production_{PF} + diffusion_{PF} - degradation_{PF} \cdot \left[PF_{div}^*\right]$$

$$\frac{d}{dt}\left[PF_{pole}^*\right] = production_{PF} + diffusion_{PF} - degradation_{PF} \cdot \left[PF_{pole}^*\right]$$

In all other cell types, these ODE are simplified to include only the diffusion and degradation of PF$_{div}$* and PF$_{pole}$*.

Last, we included a factor promoting the division of cambium* and phloem parenchyma* cells (DF*). Generally, the division chemical DF* is degraded in tissues:

$$\frac{d}{dt}\left[DF^*\right] = diffusion_{DF} - degradation_{DF} \cdot \left[DF^*\right]$$

Only, in phloem parenchyma* and cambium* cells this chemical is also produced:

$$\frac{d}{dt}\left[DF^*\right] = \frac{production_{DF} \cdot \left(\left[PF^*\right] + 100 * \left[PXY_{active}^*\right]\right)}{K + \left[PF^*\right] + 100 * \left[PXY_{active}^*\right]} + diffusion_{DF}$$

$$-degradation_{DF} \cdot \left[DF^*\right]$$

## Model 3B

In Model 3B, CLE41* is also produced in xylem cells*, such that the ODE now reads as follows:

$$\frac{d}{dt}\left[CLE41^*\right] = diffusion_{CLE41} + production_{CLE41} - \left[PXY^*\right] \cdot \left[CLE41^*\right]$$

$$-degradation_{CLE41} \cdot \left[CLE41^*\right]$$

## Model 3C

In Model 3C, the implementation of the *pxy* mutant was twofold as we needed PXY* in the model for the positional information during xylem cell* differentiation. First, the production of PXY*$_{active}$ was set to zero. Second, the DF* production only depended on DF*:

$$\frac{d}{dt}[DF^*] = \frac{production_{DF} \cdot ([PF^*])}{K + [PF^*]} + diffusion_{DF} - degradation_{DF} \cdot [DF^*]$$

## **Diffusion**

Generally, we defined the diffusion flux *phi* according to Fick's law, that is, based on the concentrations of neighboring cells and the length of the shared cell wall element

$$phi = diffusionrate \cdot length_{wallelement} \cdot \left(concentration_{cell2} - concentration_{cell1}\right)$$

so that the change in cell 1 is equal to *phi* and the change in cell 2 is equal to $-phi$. To ensure mass conservation, we included an additional factor correcting for different cell sizes:

$$\frac{d\left(concentration_{cell1}\right)}{dt} = \frac{area_{cell2}}{area_{total}} \cdot phi$$

$$\frac{d\left(concentration_{cell2}\right)}{dt} = \frac{-area_{cell1}}{area_{total}} \cdot phi$$

With $area_{total}$ defined as the sum of the sizes of cell 1 and cell 2.

In Model 1, only CLE41 diffuses between cells with no restrictions regarding to cell types. In Models 3 and 4, we also considered the diffusion of RP*, PF$_{div}$*, PF$_{pole}$*, and DF*, all of which were calculated according to the equation above and without restrictions regarding to cell types.

## Leaf.xml

A file containing the description of a tissue template as described before (*Merks et al., 2011*). The software uses this file to construct a tissue template and to run a given model.

In order to run or modify a provided model, follow the following instructions:

1. Create a new model with the desired name (e.g., 'my_cool_model') as described (*Merks et al., 2011*).
2. After a new model was created, there should be a folder '.../src/Models/my_cool_model' in your VirtualLeaf folder. In our case, the full path looked like this: 'C:\VirtualLeaf2021-main\src\ Models\ my_cool_model.'
3. In your '.../src/Models/my_cool_model' folder, locate 'my_cool_model.h' and 'my_cool_model. cpp' files. Using a text editor replace the content of those files by the content of the respective files from the model you are interested in (files provided in this article are called 'Model1.h' and 'Model1.cpp'). Please note that you should only replace the content of the files and not the files themselves. After you have completed this step, your files should still be named 'my_cool_ model.h' and 'my_cool_model.cpp.'
4. Open the files 'my_cool_model.h' and 'my_cool_model.cpp' using a text editor and replace every instance of 'Model1' by 'my_cool_model' in the text. Save the changes.
5. Locate the '.../data/leaves' folder and add the provided xml file defining the tissue template (in our case, the tissue template is called 'hypo7.xml'). The resulting full path to the file had the following structure in our case: 'C:\VirtualLeaf2021-Main\data\leaves\hypo7.xml.'

6. Compile the model as described (*Merks et al., 2011*; *Antonovici et al., 2022*). Please note that each time you introduce changes into the code, you should recompile the model and restart VirtualLeaf.

7. Now you can run VirtualLeaf. Go to the '.../bin' folder and run the 'VirtualLeaf' file. In our case, the full path looked like this: 'C:\VirtualLeaf2021-main\bin\VirtualLeaf.'

The new model will appear under the 'Models' section with the corresponding name. Please note that the name of the model that will be shown is not the same as 'my_cool_model.' Instead, it will show whichever name was indicated in the 'my_cool_model.cpp' file in this line: // specify the name of your model here; 'return QString("Model 1 – pxy only")'. In this case, there will be a new model called 'Model 1 – pxy only' in the VirtualLeaf folder under the 'Models' section.

