## [Editor Report]

The main contribution of the article is the establishment of a framework for radial plant growth that was constructed using experimental observations fed into elegant computational models. This study will be of interest to plant scientists and more generally developmental biologists, working on mechanisms of tissue growth patterning.

---

## [Decision Letter]

**Decision letter after peer review:**

Thank you for submitting your article "Computational modelling of cambium activity provides a regulatory framework for simulating radial plant growth" for consideration by *eLife*. Your article has been reviewed by 2 experts in the field, and the evaluation has been overseen by a Reviewing Editor and Naama Barkai as the Senior Editor. The following individual involved in review of your submission has agreed to reveal their identity: Przemyslaw Prusinkiewicz (Reviewer #2). The Reviewing Editor has drafted this to help you prepare a revised submission.

Major concerns rose by both reviewers relate to presentation, description, and clarification of models. Furthermore, the discussion of key model findings and their relevance should be improved. There are also major concerns on the experimental data quantification, the derived claims and how they relate to the presented models. Note this might also imply a deeper study of the presented models.

Specific comments of the reviewers:

*Reviewer #1 (Recommendations for the authors):*

Apart of the issues I have already expressed in the Public Review, please find below additional considerations:

– The authors need to work more on the clarity about some aspects of the model. How the growth is implemented in all the models? How diffusion is implemented? How the forces are implemented and the impact of the cell wall thickness? How the final timepoint is determined? I believe equation 5 needs to be revised.

Apart of the issues I have already expressed, please find below additional considerations:

– As I have already previously mentioned, an accurate image quantification with the proper replicates would help understand better the conclusions that can be extracted from the experiments. Otherwise, some statements about the results are not fully supported, and visual inspection of the images can lead to different and contradictory conclusions. For instance:

Figure 2 and Figure S1 look qualitatively different:

– In Figure 2, H4pro:mCherry seems equally expressed in the PXYpro:CFP domain and the SMLX5pro:CFP domain, without a clear maxima.

– In Figure S1, H4pro:mCherry it is much clearer its maxima at the interface of both domains, and is not that clear H4pro:mCherry is really much expression in the PXYpro:CFP domain.

Could the authors clarify these differences, and also how this is related to the statement in line 188? I would strongly recommend to do some kind of quantification, with more replicates.

The expression of PXYpro:CFP in Figure 3A looks like this marker is also reaching xylem, not just cambium. Although this seems contradictory with line 231, in which it is stated the PXYpro:CFP is surrounding the xylem cells. Quantification of the fluorescence and clarification in the text will help understand these apparent contradictions. If there is just a thin PXYpro:CFP band surrounding the xylem, it would be good to understand why this band is narrower sometimes and why is wider (eg Figure 2A)

– To have stronger conclusions about the proliferative state of the cell cycle, apart of doing the proper quantification, it might be useful to see the outline of the cells. Perhaps the combination of such data together with the modelling of the H4 marker could help to proof that cell proliferation is enhanced in that cambium region, and would enable to discard other possible interpretations.

– I would strongly suggest to study in depth the pattern formation capabilities of Model 3, including studying how the main parameters impact in the result of the outcomes (or a simplified version of it, as long as it can recapitulate the main perturbations). Perhaps this model needs to be revised, such that the cambium front instability does not occur. But perhaps the authors are in a misleading region of the parameter space. Also, the dynamics is important of these models, so there might be some observables that would be worth quantifying along time.

*Reviewer #2 (Recommendations for the authors):*

– The main weakness of this paper is, in my opinion, the presentation of the results. The progression through a sequence of models has some rationale – it presumably depicts the path of refinements through which the authors arrived at the final model – but makes this paper tedious to read and obscures key results. It would be beneficial if the paper focused on a well-organized, crisp, mathematically sound presentation of the final model, including all relevant equations, put carefully in the context of the essential molecular-level information, and with clearly stated discussion of lessons learned and shortcomings of the model. The intermediate steps, which may be of interest to researchers who want to continue this line of study, could be delegated to supplementary materials.

– Summary of key molecular players in lines 76-105 is very dense and requires an explanatory figure illustrating the proposed interactions.

– Although ARF5 is mentioned early on (line 101), the role of auxin is downplayed (c.f. lines 441-443). However, the recent paper by Smetana et al. (ref. 10) attributes a significant role to auxin. This discrepancy should at least be discussed.

– The authors tacitly assume that considering the cross-section of an organ such as the hypocotyl suffices to explain the radial organization of its tissues. Perhaps this is true; nevertheless, it is an assumption, which should be clearly stated as such. In particular, if auxin indeed is involved in the tissue patterning, wouldn't its flow in the longitudinal direction play a role?

– There are previous modeling papers dealing with the organization of tissues in shoots and roots, in particular:

Marta Ibañes et al., Brassinosteroid signaling and auxin transport are required to establish the periodic pattern of Arabidopsis shoot vascular bundles, PNAS 2009

N. Fàbregas et al., Auxin Influx Carriers Control Vascular Patterning and Xylem Differentiation in *Arabidopsis thaliana*, PLOS Genetics 2015

These papers should be cited.

– The idea of distinguishing real substances, tissues etc. from their models using an asterisk is an interesting one; however, in order to be really helpful it should be applied very consistently throughout. Currently there are inconsistencies: for example, in the legend to the right of Figure 5F symbol DF is not starred, but in the figure caption it is.

– The systems of equations representing the models are not adequately presented. For example, Model 1 deals with xylem, phloem and cambium cells, the latter in two states (Figure 1), yet Equations 2-4 apparently characterize only cambium cells. Further models are described even less rigorously. The list of parameters (Table S1) is obscure, as the equations involving listed parameters are not explicitly given. The readers should not be required to reverse-engineer the code to understand how the models really work.

– The models describe a growing system, yet the equations do not include a term representing a decrease in concentrations of molecules in the cells due to cell growth. Why?

– There are also further questions regarding the models. How and on what basis was the initial tissue template (Figure 1A) specified? What are the initial values of the variables? What are the mathematical conditions for attributing types to cells in the tissue?

– Why it the initial frame ( = the template?) in Model 3B (Movies 5x) different from those used previously?

– The table in Figure 5 G includes the heading "Initial values (after run)": What does this mean? (It sounds like an oxymoron.)

– Hollow phrases such as "minimal framework of intercellular communication loops" (lines 37-38) or "reciprocal and interconnected gradients of regulators along the radial sequence of tissues" (line 483) should be avoided, especially in the absence of a proper description of the mathematical models.

– The section on the physical properties of cambium does not adequately address the problem of radial file formation. The question of factors determining the orientation of cell divisions has been extensively addressed in literature (including numerous models), and references to three old papers do not sufficiently position the issue. The algorithm employed in the models to determine the orientation of the divisions is not explained (presumably, the authors just rely on a default algorithm implemented in Virtual Leaf). The observation that stiffening of xylem cell walls may contribute to the organization of cells into files is of some interest, but begs a biomechanical analysis, which is absent.

– Lines 435-437. Listing Turing (reference 43) as representative of "using positional information mediated by morphogenetic gradients" is inappropriate, given the long history of tension between Turing's reaction-diffusion concept and Wolpert's idea of positional information. The paper by J. Green and J. Sharpe, "Positional information and reaction-diffusion: two big ideas in developmental biology combine", Development 2015, provides an authoritative recent perspective on this tension.

– The discussion seems to be largely disconnected from the content of the paper. In particular, the relation of the text beginning in line 449 to the modeling effort is not clear. Is the phrasing "In contrast to our expectations…" supposed to mean "In contrast to the model"? Then, what is the reason for the discrepancy? Or, were these expectations independent of the model, in which case the question arises, how are they related to the topic of the paper, "Computational modelling of cambium activity…"

– Lines 579-580 state: "All simulations within Model 1, Model 2, and Model 3 […] were repeated at least ten times." Why? If the models were deterministic, they would produce the same results. If they included stochastic terms, the results of different runs would differ, but there is no indication of stochastic terms in the paper.

– Line 592 and following: Which parameters were optimized? The statement "The parameter space contains an interval for each parameter from which the parameter value can be chosen" is unclear: what would be a parameter for which a value cannot be chosen? Also, how many runs were performed to optimize? How was the end of the optimization process decided?

[Editors' note: further revisions were suggested prior to acceptance, as described below.]

Thank you for resubmitting your work entitled "Computational modelling of cambium activity provides a regulatory framework for simulating radial plant growth" for further consideration by *eLife*. Your revised article has been evaluated by Naama Barkai (Senior Editor) and a Reviewing Editor.

The manuscript has been improved but there are some remaining issues that need to be addressed, as outlined below:

Most of the remaining issues relate to the either lack of detailed explanations or confusing sentences. I suggest authors go carefully through all comments of Reviewers and try to address those for clarification.

The authors should also provide a quantitative analysis of radial fluorescence profiles as they previously performed in Shi et al. 2019. I believe this analysis will further strengthen authors claims. For ease, there is a number of alternative strategies that Reviewer #1 suggested for authors to consider while doing this analysis.

*Reviewer #1 (Recommendations for the authors):*

I appreciate the replies of the authors to my comments and the points they addressed, I think the quality of the manuscript has improved. See below some other comments I would like to make of this new revised version of the manuscript.

– I still find some statements about the interpretation of the cambium activity fluorescence reporters might require to be better supported. I understand that a detailed quantification might not be possible, but I would strongly recommend authors to do radial intensity profiles to support their statements, as the authors very nicely did in a previous publication [Shi et al. 2019] (e.g. one could do it with Fiji, with a certain line width and doing some binning along the line to smoothen fluctuations).

For instance, these are interpretations of the authors I did not find convincing:

Lines 284-287: 'PXY promoter reporter activity was observed distally to xylem sectors, whereas the SMXL5 promoter activity was as usual present distally to the PXY activity domain. Interestingly, PXYpro:CFP and SMXL5pro:YFP activity domains were still completely distinct '

I would say PXY promoter activity was observed also within the xylem sectors. I am not sure if PXYpro:CFP and SMXL5pro:YFP would be indeed completely distinct, perhaps they overlap?

It would be good to show the radial quantified profiles in WT as well, to better appreciate the differences with the mutants.

If the authors still find quantification is not the way to go, I would suggest doing zooms of the regions of interest, and showing single and composite channels to facilitate the interpretation.

– In the IRX3pro:CLE41 mutant, the statement about lower number of xylem cells should be better supported; looking at the time course of Figure 3—figure supplement 2, earlier time points in the IRX3pro:CLE41 line might suggest this is not the case. I think this raises the question of having more repeats to support this statement (having said that, more repeats of the pxy mutant would be also desirable). Also, I am wondering whether authors are referring to absolute numbers of xylem cells or fraction of xylem cells, could they clarify? Also, to support the claimed statements, a quantification with ilastik of xylem cells might be realistic to do.

– The authors conclude that the cell wall thickness in the procambial cells is smaller than in its surrounding tissues. The way they show it should be revised. First, it is not clear if they do it with Direct Red 23 (line 418) or Direct Yellow 96 (mentioned in the caption of Figure4—figure supplement 1). Second, it is not clear the mean intensity of Direct Red 23 or Direct Yellow could be a good proxy of cell wall thickness – could the authors justify this? (I am not an expert in this topic, but this should be clear and justified to non-experts as myself). In the case of Direct Yellow 96, the mean intensity might be related to the amount of xyloglucans if I understand it well from Ursache et al. 2018; in the case of Direct Red 23, I understand the fluorescence is related to cellulose content at a given part of the cell wall – but not forcely to thickness, and therefore overall stiffness. Third, the quantification using the Radial Profile function might be very misleading, given there can be other factors affecting the outcome, such as the density of cells at a given radial binning, the cell heterogeneity while being a tissue averaged measure, etc – better to do it at a cellular resolution.

– The diffusion operator as described in the appendix, if I understood it well and I am not wrong, it would not fulfil conservation of mass if it is applied to a tissue made of cells of different sizes. For instance, if you apply your operator to two cells with very two different sizes and make the numbers in terms of the exchange number of molecules (i.e., convert the expression of concentrations to number of molecules and cell volumes), the larger cell will have more flux of molecules than the smaller cell. Given the modelled tissue has cell size heterogeneity that can not be avoided, why didn't the authors use laplacians that could follow a conservation of mass such as in Sukumar and Bolander (2003)? I am wondering whether this violation of mass conservation might affect the presented computational results.

– I appreciate the performed model robustness analysis by the authors. For completeness, I think it would be important to include some additional simulations assessing the effect of diffusion rates (and/or the degradation of the modelled diffusible species), at least for CLE41. I believe some parameter ranges might stabilize the front – my suggestion is not to push to an in depth study on the front instability, which I understood is beyond the scope of this manuscript, but rather to ask for a more complete and robust analysis of essential parameters of the model.

---

## [Author Response]

Reviewer #1 (Recommendations for the authors):Apart of the issues I have already expressed in the Public Review, please find below additional considerations:– The authors need to work more on the clarity about some aspects of the model. How the growth is implemented in all the models? How diffusion is implemented? How the forces are implemented and the impact of the cell wall thickness? How the final timepoint is determined? I believe equation 5 needs to be revised.Apart of the issues I have already expressed, please find below additional considerations:

We have provided an expanded description of the simulations in VirtualLeaf in the supporting information including the implementation of growth and diffusion as well as the forces. We have included a detailed description of the changes to the source code of VirtualLeaf for Model 3D and illustrated the impact of the cell wall thickness on the simulations. Simulations were run until central features of the tissue were represented by the model – bidirectional tissue formation, positional cell differentiation and distribution of phloem poles. We thank the reviewer for pointing out the issue with equation 6 (previously equation 5), which has now been revised to list the stimulating chemicals as well as their contribution to DF* production.

– As I have already previously mentioned, an accurate image quantification with the proper replicates would help understand better the conclusions that can be extracted from the experiments. Otherwise, some statements about the results are not fully supported, and visual inspection of the images can lead to different and contradictory conclusions. For instance:Figure 2 and Figure S1 look qualitatively different:– In Figure 2, H4pro:mCherry seems equally expressed in the PXYpro:CFP domain and the SMLX5pro:CFP domain, without a clear maxima.– In Figure S1, H4pro:mCherry it is much clearer its maxima at the interface of both domains, and is not that clear H4pro:mCherry is really much expression in the PXYpro:CFP domain.Could the authors clarify these differences, and also how this is related to the statement in line 188? I would strongly recommend to do some kind of quantification, with more replicates.

As mentioned below, we agree with these concerns toward the H4 marker used in the initial submission. Because H4 expression is not specifically associated with cell division but with DNA synthesis in general and, thus, with endoreduplication, H4 expression does not report faithfully on cell division. As a response, we removed related figures and now reference our previous study characterizing cell division levels in different cambium domains based on cell linage analyses (Shi et al., 2019). Because this is a far more reliable analysis and convincingly supports our claims, we believe that we thereby addressed this concern. Quantification of images is difficult otherwise due to the stochastic (non‐quantitative) nature of anatomical defects in *pxy* and IRX3:CLE41 lines.

The expression of PXYpro:CFP in Figure 3A looks like this marker is also reaching xylem, not just cambium. Although this seems contradictory with line 231, in which it is stated the PXYpro:CFP is surrounding the xylem cells. Quantification of the fluorescence and clarification in the text will help understand these apparent contradictions. If there is just a thin PXYpro:CFP band surrounding the xylem, it would be good to understand why this band is narrower sometimes and why is wider (eg Figure 2A)

We agree with the reviewer that this statement was misleading. As reported previously (Shi et al., 2019), PXY promoter activity is not only found in proximal cambium cells but also in the xylem tissue itself. We revised this statement accordingly.

– To have stronger conclusions about the proliferative state of the cell cycle, apart of doing the proper quantification, it might be useful to see the outline of the cells. Perhaps the combination of such data together with the modelling of the H4 marker could help to proof that cell proliferation is enhanced in that cambium region, and would enable to discard other possible interpretations.

See our comments above.

– I would strongly suggest to study in depth the pattern formation capabilities of Model 3, including studying how the main parameters impact in the result of the outcomes (or a simplified version of it, as long as it can recapitulate the main perturbations). Perhaps this model needs to be revised, such that the cambium front instability does not occur. But perhaps the authors are in a misleading region of the parameter space. Also, the dynamics is important of these models, so there might be some observables that would be worth quantifying along time.

We thank the reviewers for suggesting a more throughout parameter analysis. We have now re‐run the parameter search until we obtained 5 distinct parameter sets that recapitulate the central features of cambium activity to ensure that the behavior of the model is robust across the analyzed parameter space. We further implemented experimentally determined ratios for cell wall stabilities in phloem, cambium and xylem cells in an effort to improve the model. This has made the xylem region more compact – and in turn also influenced the shape of the cambium around it. However, this did not solve the issue of the edge instability suggesting that other mechanisms are at play here that are beyond the CEL41‐PXY module and simple mechanic mechanisms like cell type specific cell wall stability values. As we do not know how front stability is achieved in planta and, therefore, do not know how to implement this in the computational model, analyzing front instability is currently beyond the scope of our computational modelling approach but a very interesting aspect nevertheless which is now discussed in the Discussion section.

Reviewer #2 (Recommendations for the authors):– The main weakness of this paper is, in my opinion, the presentation of the results. The progression through a sequence of models has some rationale – it presumably depicts the path of refinements through which the authors arrived at the final model – but makes this paper tedious to read and obscures key results. It would be beneficial if the paper focused on a well-organized, crisp, mathematically sound presentation of the final model, including all relevant equations, put carefully in the context of the essential molecular-level information, and with clearly stated discussion of lessons learned and shortcomings of the model. The intermediate steps, which may be of interest to researchers who want to continue this line of study, could be delegated to supplementary materials.

(See our comments above) We completely see the point of this suggestion. We discussed the possibility to immediately show Model 3 among the authors but the general opinion was that going through Model 1 and 2 first, will help the readers tremendously to follow why Model 3 is designed as it is. As our aim was to identify the minimal framework around the CLE41/PXY signaling module, we started with the most simplistic model possible and develop Model 3 from there by iterative comparisons of in planta and in silico tissue patterns generating, in our view, a quite attractive storyline. To better explain Model 3, we added all equations and a more detailed description in the main text and the supplemental information.

– Summary of key molecular players in lines 76-105 is very dense and requires an explanatory figure illustrating the proposed interactions.

A large body of literature describes these interactions over many years now, which we also cite. Because we want to avoid the impression that we intend to claim that we represent some of the described factors (e.g. WOX4, BES1, …) by our model, we would abstain from such an explanatory scheme as it, in our opinion, would restrict the readers’ view to look for respective representatives in our model.

– Although ARF5 is mentioned early on (line 101), the role of auxin is downplayed (c.f. lines 441-443). However, the recent paper by Smetana et al. (ref. 10) attributes a significant role to auxin. This discrepancy should at least be discussed.

Thanks for this comment. We discuss the issue of not including the role of auxin and the ‘organizing center’ identified in Smetana et al. in the discussion part.

– The authors tacitly assume that considering the cross-section of an organ such as the hypocotyl suffices to explain the radial organization of its tissues. Perhaps this is true; nevertheless, it is an assumption, which should be clearly stated as such. In particular, if auxin indeed is involved in the tissue patterning, wouldn't its flow in the longitudinal direction play a role?

We agree, that longitudinal signaling is one central aspect of our biological system which we were not able to consider in our simulations, due to the immense effort this would require. This aspect is now mentioned in the discussion.

– There are previous modeling papers dealing with the organization of tissues in shoots and roots, in particular:Marta Ibañes et al., Brassinosteroid signaling and auxin transport are required to establish the periodic pattern of Arabidopsis shoot vascular bundles, PNAS 2009N. Fàbregas et al., Auxin Influx Carriers Control Vascular Patterning and Xylem Differentiation in *Arabidopsis thaliana*, PLOS Genetics 2015These papers should be cited.

These papers are now cited in the discussion part.

– The idea of distinguishing real substances, tissues etc. from their models using an asterisk is an interesting one; however, in order to be really helpful it should be applied very consistently throughout. Currently there are inconsistencies: for example, in the legend to the right of Figure 5F symbol DF is not starred, but in the figure caption it is.

Thanks for the remark. This has been corrected.

– The systems of equations representing the models are not adequately presented. For example, Model 1 deals with xylem, phloem and cambium cells, the latter in two states (Figure 1), yet Equations 2-4 apparently characterize only cambium cells. Further models are described even less rigorously. The list of parameters (Table S1) is obscure, as the equations involving listed parameters are not explicitly given. The readers should not be required to reverse-engineer the code to understand how the models really work.

Again, thanks for raising this important point. We have now included an entire list of the equations for all model versions and all cell types and more detailed explanations of the models (supplemental methods). Additionally, we have expanded Figure 1 to include a schematic representation of the model structure, where cell‐type specific reactions are colored accordingly. We feel that listing all equations for all cell types in the main text of the manuscript would disrupt the reading flow too much. Also, parameters are now explicitly depicted in Supplementary File 2 (former Table S1). We have also enclosed a table with all conditions for cell differentiation processes for all model structures (Supplementary File 1).

– The models describe a growing system, yet the equations do not include a term representing a decrease in concentrations of molecules in the cells due to cell growth. Why?

The equations for all model species include a degradation term, which means that a steady state between production, diffusion and degradation is reached. As these processes occur on a faster timescale that cell growth, we felt that including a factor describing the dilution by cell growth would provide little improvement on the simulations while notably increasing the computational costs for each simulation.

– There are also further questions regarding the models. How and on what basis was the initial tissue template (Figure 1A) specified? What are the initial values of the variables? What are the mathematical conditions for attributing types to cells in the tissue?

The initial tissue template was designed based on a typical Arabidopsis cross section at the onset of radial growth with some simplifications which we mention in the discussion part. We have now listed the conditions for all cell differentiations in the supporting information as part of a more elaborate description of the models.

– Why it the initial frame ( = the template?) in Model 3B (Movies 5x) different from those used previously?

We have re‐run all simulations with the new version of VirtualLeaf and used in all cases the identical starting frame.

– The table in Figure 5 G includes the heading "Initial values (after run)": What does this mean? (It sounds like an oxymoron.)

We have now expanded the parameter search, added information on the parameter values and fitted model behavior to the supporting information and designed a new Figure 5G. The heading “initial values (after run)” was indeed an oxymoron – we were referring to the cell type proportions at the end of the simulation using the initial parameter set of model 3A. This is now included in the new Figure 5G but named “Model 3A”.

– Hollow phrases such as "minimal framework of intercellular communication loops" (lines 37-38) or "reciprocal and interconnected gradients of regulators along the radial sequence of tissues" (line 483) should be avoided, especially in the absence of a proper description of the mathematical models.

Thanks for this comment. We changed or removed these statements.

– The section on the physical properties of cambium does not adequately address the problem of radial file formation. The question of factors determining the orientation of cell divisions has been extensively addressed in literature (including numerous models), and references to three old papers do not sufficiently position the issue. The algorithm employed in the models to determine the orientation of the divisions is not explained (presumably, the authors just rely on a default algorithm implemented in Virtual Leaf). The observation that stiffening of xylem cell walls may contribute to the organization of cells into files is of some interest, but begs a biomechanical analysis, which is absent.

We agree with the reviewer that a biomechanical analysis would nicely complement our study. In fact, we are doing such an analysis at the moment which, however, will require 2‐3 additional years to be completed and is thus beyond the scope of this work. The point of including a computational approach in this regard in our manuscript is to highlight the potential for our tool to address mechanical aspects of plant growth and to generate related hypotheses which was not possible using the VirtualLeaf platform before. We believe that this will be very useful for readers being interested in this aspect and the stepping stone for further analyses, let it be on the computation and/or the biomechanical side. We included am extended description of our models in the supplemental information which we hope is helpful in this context.

– Lines 435-437. Listing Turing (reference 43) as representative of "using positional information mediated by morphogenetic gradients" is inappropriate, given the long history of tension between Turing's reaction-diffusion concept and Wolpert's idea of positional information. The paper by J. Green and J. Sharpe, "Positional information and reaction-diffusion: two big ideas in developmental biology combine", Development 2015, provides an authoritative recent perspective on this tension.

Again, thanks for this remark. We replaced the original citations with the suggested one and indicated the controversial nature of this concept.

– The discussion seems to be largely disconnected from the content of the paper. In particular, the relation of the text beginning in line 449 to the modeling effort is not clear. Is the phrasing "In contrast to our expectations…" supposed to mean "In contrast to the model"? Then, what is the reason for the discrepancy? Or, were these expectations independent of the model, in which case the question arises, how are they related to the topic of the paper, "Computational modelling of cambium activity…"

We would politely disagree with the statement that the discussion part is largely disconnected from the content of the paper. Still, we agree that the mentioned phrase was ambiguous. In this paragraph we deal with the discrepancy of our results (Figure 3G‐I) with the long‐standing view on the role of PXY in xylem formation and not so much with our modelling efforts. We specified this part accordingly.

– Lines 579-580 state: "All simulations within Model 1, Model 2, and Model 3 […] were repeated at least ten times." Why? If the models were deterministic, they would produce the same results. If they included stochastic terms, the results of different runs would differ, but there is no indication of stochastic terms in the paper.

During tissue simulations in VirtualLeaf, all nodes are moved in a random order, which can affect cell sizes (and with that the timing of cell differentiation) and cellular dimensions (which would influence the direction of cell divisions). To ensure that the model simulations are robust with respect to these fluctuations, we ran repetitions for all simulations in all models.

– Line 592 and following: Which parameters were optimized? The statement "The parameter space contains an interval for each parameter from which the parameter value can be chosen" is unclear: what would be a parameter for which a value cannot be chosen? Also, how many runs were performed to optimize? How was the end of the optimization process decided?

In comparison to the first version, we considerably extended our parameter search. We optimized the parameters, which are most closely related to tissue formation: cell differentiation thresholds, cell division thresholds and maximal cell sizes. All of these parameters, including a brief description of the parameter, are listed in Supplementary File 2. As Model 3A was already able to describe central features of cambium activity, we limited the parameter search space to 1/3‐ and 3‐ times the original parameter value. In total 5,000 runs were performed to explore the parameter space. The final simulation results were then evaluated based on the cell type proportions. We stopped the parameter search, once we had obtained five distinct parameter sets, which can reproduce the central features of the tissue morphology. This procedure is explained in the Materials and methods section.

[Editors' note: further revisions were suggested prior to acceptance, as described below.]

Reviewer #1 (Recommendations for the authors):I appreciate the replies of the authors to my comments and the points they addressed, I think the quality of the manuscript has improved. See below some other comments I would like to make of this new revised version of the manuscript.– I still find some statements about the interpretation of the cambium activity fluorescence reporters might require to be better supported. I understand that a detailed quantification might not be possible, but I would strongly recommend authors to do radial intensity profiles to support their statements, as the authors very nicely did in a previous publication [Shi et al. 2019] (e.g. one could do it with Fiji, with a certain line width and doing some binning along the line to smoothen fluctuations).For instance, these are interpretations of the authors I did not find convincing:Lines 284-287: 'PXY promoter reporter activity was observed distally to xylem sectors, whereas the SMXL5 promoter activity was as usual present distally to the PXY activity domain. Interestingly, PXYpro:CFP and SMXL5pro:YFP activity domains were still completely distinct 'I would say PXY promoter activity was observed also within the xylem sectors. I am not sure if PXYpro:CFP and SMXL5pro:YFP would be indeed completely distinct, perhaps they overlap?

Thanks for this comment. We agree that the wording was misleading in this case. As a response, we deleted this sentence as is was also redundant with the sentence before and adapted the text accordingly. The text now reads: “In pxy mutants, the xylem tissue did not have a cylindrical shape, but was instead clustered in radial sectors showing PXYpro:CFP and, at their distal ends, SMXL5pro:YFP activity, whereas regions in between those sectors had little to no xylem and did not show reporter activity (Figure 3E, Figure 2—figure supplement 1, Figure 3—figure supplement 1, Figure 3—figure supplement 2).”

It would be good to show the radial quantified profiles in WT as well, to better appreciate the differences with the mutants.If the authors still find quantification is not the way to go, I would suggest doing zooms of the regions of interest, and showing single and composite channels to facilitate the interpretation.

Again, thanks for this comment. We tried to perform radial profiling of reporter activities in the different backgrounds but, due to anatomical differences in pxy mutants and in IRX3pro:CLE41 plants, signal noise between samples was too large to generate interpretable results. Instead, we now provide three examples of close‐ups of the cambium domain in wild type and pxy mutants (Figure 3‐figure supplement 3). Careful analysis of these images resulted in a modification of our previous statement mentioned above and now reads: “Interestingly, PXYpro:CFP and SMXL5pro:YFP activity domains were still mostly distinct meaning that PXYpro:CFP activity did not expand further beyond established xylem than in wild type (Figure 3E, Figure 3—figure supplement 3). This discrepancy indicated that, in contrast to our assumption, the CLE41‐PXY signaling module did not restrict PXY promoter activity in the distal cambium. Of note, the sharp border between PXYpro:CFP and SMXL5pro:YFP activity was less pronounced in pxy mutants mostly due to a spread of SMXL5pro:YFP activity towards xylem tissues (Figure 3—figure supplement 3).”

– In the IRX3pro:CLE41 mutant, the statement about lower number of xylem cells should be better supported; looking at the time course of Figure 3—figure supplement 2, earlier time points in the IRX3pro:CLE41 line might suggest this is not the case. I think this raises the question of having more repeats to support this statement (having said that, more repeats of the pxy mutant would be also desirable). Also, I am wondering whether authors are referring to absolute numbers of xylem cells or fraction of xylem cells, could they clarify? Also, to support the claimed statements, a quantification with ilastik of xylem cells might be realistic to do.

Thanks for pointing out our imprecise wording. As one response, we added another comparison between wild type, IRX3 plants and pxy mutants (Figure 3‐figure supplement 2). As another response we exchanged the term ‘xylem cells’ by the more correct term ‘vessel elements’ as those are prominently stained by Direct Red 23 and, thus, are the only xylem‐related cell type which can be non‐ambiguously identified. We did not use this term before because we only speak about ‘xylem cells’ in the context of our model and, for non‐familiar readers, we had the feeling that this may be confusing. As the reviewer will see, the second example confirms a reduced density in vessel elements in IRX3 lines (not in differentiated xylem cell number as previously stated). This is what we now state in the text: “PXYpro:CFP activity was found in irregularly shaped patches containing differentiated xylem vessel elements distributed over the whole cross‐section.”

– The authors conclude that the cell wall thickness in the procambial cells is smaller than in its surrounding tissues. The way they show it should be revised. First, it is not clear if they do it with Direct Red 23 (line 418) or Direct Yellow 96 (mentioned in the caption of Figure4—figure supplement 1). Second, it is not clear the mean intensity of Direct Red 23 or Direct Yellow could be a good proxy of cell wall thickness – could the authors justify this? (I am not an expert in this topic, but this should be clear and justified to non-experts as myself). In the case of Direct Yellow 96, the mean intensity might be related to the amount of xyloglucans if I understand it well from Ursache et al. 2018; in the case of Direct Red 23, I understand the fluorescence is related to cellulose content at a given part of the cell wall – but not forcely to thickness, and therefore overall stiffness. Third, the quantification using the Radial Profile function might be very misleading, given there can be other factors affecting the outcome, such as the density of cells at a given radial binning, the cell heterogeneity while being a tissue averaged measure, etc – better to do it at a cellular resolution.

We fully understand the motivation for this comment. As a response, we first changed ‘Direct Red 23’ to ‘Direct Yellow 96’ in the main text as this was indeed incorrect. As a justification for using Direct Yellow 96 as a proxy for cell wall thickness, we’d like to argue that Direct Red 23 and Direct Yellow 96 behave similarly in stainings of root tips which also includes xylem vessels containing secondary cell walls (Ursache et al., 2018). Moreover, also Direct Red 23 application leads to the same relative staining levels as Direct Yellow 96 when comparing cambium stem cells and surrounding cells, a result which is now included in Figure 4—figure supplement 2. Because Direct Red 23 is indeed considered as a cellulose stain (references in Ursache et al., 2018), we believe that using Direct Yellow 96 staining as a proxy for cell wall thickness (i.e. stiffness) is justified. When quantifying Direct Yellow 96 staining for each cell which we performed for a different study (see included picture), the reviewer will see that differences between cambium stem cells and other cells are quite pronounced in some cases and much less in other cases, making it difficult to choose a specific ratio based on cell‐specific measurements. Our choice of 50% stiffness of cambium stem cells in comparison to surrounding cells is also supported by Atomic Force Microscopy of cell walls being present in the cambium region (Arnould et al., 2022). In this study, tension wood is analyzed with very high spatial resolution also revealing the different layer within the cell walls. Ignoring G‐layers which is specific for tension wood, indentation modulus is ~5 GPa (S1+CML) for cambium cell walls and ~10 GPa for differentiated cells (S2). Taken these points together, we prefer to stick to the original approach and use our previous calculation as an approximation for determining a ration between the stiffness of cambium stem cells and surrounding cells. The main text references now the respective analysis.

– The diffusion operator as described in the appendix, if I understood it well and I am not wrong, it would not fulfil conservation of mass if it is applied to a tissue made of cells of different sizes. For instance, if you apply your operator to two cells with very two different sizes and make the numbers in terms of the exchange number of molecules (i.e., convert the expression of concentrations to number of molecules and cell volumes), the larger cell will have more flux of molecules than the smaller cell. Given the modelled tissue has cell size heterogeneity that can not be avoided, why didn't the authors use laplacians that could follow a conservation of mass such as in Sukumar and Bolander (2003)? I am wondering whether this violation of mass conservation might affect the presented computational results.

Thank you for pointing this out. We have now included a scaling factor in the diffusion between cells that ensures mass conservation for these equations. Here, we decided to work with what had already been programmed in VirtualLeaf and chose a scaling factor that is based on the cells’ area. An explanation of this factor is also included in the supplementary material on simulations in VirtualLeaf. We implemented this correction for all models (Model 1 to Model 4) and we have repeated all simulations, including the parameter search. While small differences between the previous simulations can be observed, the qualitative behavior remains the same.

– I appreciate the performed model robustness analysis by the authors. For completeness, I think it would be important to include some additional simulations assessing the effect of diffusion rates (and/or the degradation of the modelled diffusible species), at least for CLE41. I believe some parameter ranges might stabilize the front – my suggestion is not to push to an in depth study on the front instability, which I understood is beyond the scope of this manuscript, but rather to ask for a more complete and robust analysis of essential parameters of the model.

Thank you for this comment. We have now expanded the parameter search to include the diffusion and degradation rate for CLE41 after implementing the correction to ensure mass conservation. While it seems that some parameter ranges result in more edge instability, it does seem like other mechanisms are at play here that are beyond the CLE41‐PXY signaling module.